**Shoreline and Land Use Land Cover Changes along the 2004 tsunami-**
**affected South Andaman Coast: Understanding Changing Hazard**
**Susceptibility**
Vikas Ghadamode[1,2], Aruna Kumari Kondarathi [1], Anand K Pandey[1,2,§] Kirti Srivastava[1]
1. CSIR- National Geophysical Research Institute, Hyderabad, 500007 India.
2. Academy of Scientific and Innovative Research (AcSIR), Ghaziabad 201002, India.
§ Corresponding author: Email address: akpandey@ngri.res.in
*Tel: +91-40-27012416*
**Abstract**
The 2004 tsunami affected the South Andaman coast, experiencing dynamic changes in the
coastal geomorphology, making the region vulnerable. We focus on pre-and post-tsunami
shoreline and Land Use Land Cover changes for 2004, 2005, and 2022 to analyse the dynamic
change in hazard. We used GEBCO bathymetry data to calculate Run-up (m), arrival times
(Min), and inundation (m) at a few locations using three tsunamigenic earthquake source
parameters, namely the 2004-Sumatra, 1941-North Andaman, and 1881-Car Nicobar
earthquakes. The Digital Shoreline Analysis System is used for the shoreline change estimates.
The Landsat data is used to calculate shoreline and Land Use Land Cover (LULC) change in
five classes, namely Built-Up Areas, Forests, Inundation areas, Croplands, and water bodies
during the above period. We examine the correlation between the LULC changes and the
dynamic change in shoreline due to population flux, infrastructural growth, and Gross State
Domestic Product growth. India industry estimates the Andaman & Nicobar Islands losses
exceed INR 10 billion during 2004, which would see a five-fold increase in economic loss due
to a doubling of built-up area, a three-fold increase in tourist inflow, and a population density
growth. The unsustainable decline in the forest cover, mangroves, and cropland would affect
sustainability during a disaster despite coastal safety measures.
**Keywords: Geomorphology, Land use Land cover, Shoreline, Tsunami, Remote sensing**

**1. Introduction:**

The Coastal shorelines are dynamic and highly vulnerable to erosion and accretion caused by hydrodynamic, tectonic, geomorphic, and climate forcing, including tsunamis, cyclones, flooding, storm surges, wave action, wind and tide changes, and sea level variations (Nayak 2002; Boak &Turner 2005; Kumar et al., 2010; Mukhopadhyay et al., 2011). In addition to natural coastal processes, coastal resources are constantly under stress due to anthropogenic activities, such as industrialization, port construction, beach sand mining, garbage dumping, urbanization, trade, tourism, and recreational activities, which significantly impact the shoreline and results into damage to natural ecosystems (Yi et al., 2018; Davis, 2019). It is important to regularly monitor spatiotemporal along shorelines, Land use / Land Cover (LULC), and geomorphic features (Moran, 2003; Cooper et al., 2004; Scheffers et al., 2005; Jayakumar & Malarvannan, 2016). Several studies have analyzed various coastal processes, including mapping shoreline change, LULC change detection, and analysis of geomorphological landforms using satellite data. The temporal multispectral satellite data allow for the identification of regions undergoing erosion or accretion change (Misra and Balaji, 2015; Kumari et al., 2012; Tonisso et al., 2012; Murali et al., 2013; Sudha Rani et al., 2015; Rowland et al., 2022; Thiéblemont et al., 2021). The M 9.3 undersea earthquake on December 26, 2004, near the coast of Sumatra, Indonesia, triggered the Indian Ocean tsunami and caused massive destruction of the coastal ecosystem in the Andaman region (Sheth et al., 2006; Ramalanjaona, 2011). Several researchers analyzed shoreline and geomorphological changes of the 2004 Sumatra tsunami using remote sensing data (Kumari et al., 2012; Yuvaraj et al., 2014; Yunus and Narayana, 2015; Yunus et al., 2016).

Since the 2004 tsunami, the Andaman and Nicobar Islands have experienced notable population growth, infrastructural development, and flourishing tourism activities over the past decade (Yuvaraj et al., 2014). The development is profound in the south Andaman region. This

is a cause of concern for the tsunami vulnerability as the region is prone to large earthquakes
and is a seismo-tectonically active plate boundary. In this study, we Compute Tsunami arrival
times, run-up heights, and inundation extent along the south Andaman region. We also
analyzed dynamic vulnerability using temporal and spatial changes in shoreline and LULC for
the tsunami-affected areas (Velmurugan et al., 2006; Ghadamode et al., 2022). The analysis
covers three time periods: 2004 (pre-tsunami), 2005 (post-tsunami), and 2022 (current state) of
shoreline changes using multi-temporal Landsat data employing the End Point Rate (EPR) and
Net Shoreline Movement (NSM) methods (Himmelstoss et al., 2021) and LULC changes. A
relationship between LULC changes and vital socioeconomic factors such as population
dynamics, tourism trends, and the Gross State Domestic Product (GSDP) is established to
assess the potential future impacts of tsunamis in the region. The results would provide
actionable insights to the policymakers, coastal planners, and stakeholders in disaster
management and sustainable coastal development.
**2. Study Area**
South Andaman region, with ~1,262 km² area and a 413 km coastline, is the
southernmost island of the Great Andaman, where most of the Andaman Island's population
and infrastructure are centered. As per the 2011 Indian census, South Andaman has a
population of 238,142 people, which increased to 266,900 in 2021 (estimate based on
www.census2011.co.in). The most habitable areas in the eastern part of South Andaman are
located on low lands at bay heads in addition to the higher slopes bordering bays and coastal
flat lands (Ghosh et al., 2004), which experienced devastation and losses during the 2004
Tsunami (Fig. 1). We selected 13 locations, namely South Point in Port Blair, Rutland Island,
Corbyn's Cove Beach, Madhuban Bay, Brichgunj, Chidiyatopu, Thirupatti Temple,
Wandoorjetty, Bamboo Flat, Potatang, Shoal Bay, Radha Nagar, and Govinda Nagar (Fig. 1)
for vulnerability assessment in the present study.

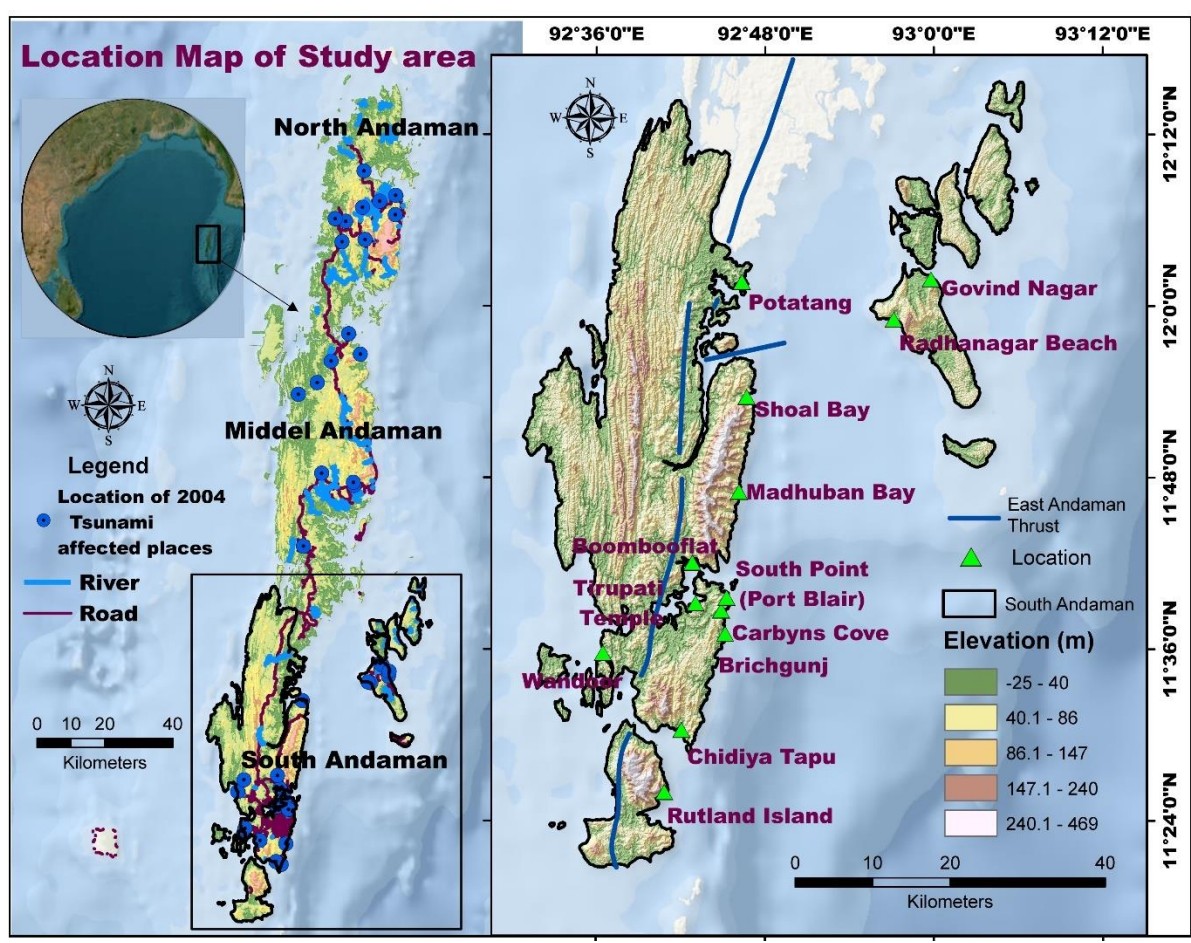

*Figure 1 Location Map of the South Andaman Region (© Google Maps & © Google Earth).*

83        The tectonic activity and weathering processes have influenced the region's topography

growth and evolution (Curray, 2005; Bandopadhyay and Carter, 2017). The East Andaman
Thrust, also called East Boundary Thrust, is a linear/curvilinear ~500 km long fault zone and
is the locus of ongoing convergent and crustal deformation along the Sunda-Andaman plate
boundary. This structure is pivotal in creating accretionary prisms within the outer-arc ridge of
the Andaman and Nicobar subduction zones (Fig. 1; Bhat et al., 2023).

89        The structure-bound major geomorphological features in South Andaman include hills,

valleys, beaches, mangroves, and coral reefs (Fig. 2a). The highest peak on the island is Mount
Harriet, with approximately 1,200 m (3,937 feet) (southandaman.nic.in). The north-western
and north-eastern parts of South Andaman are highly and moderately dissected, whereas the
Southern part has low dissected structural hills and valleys (Fig. 2a, b, c, and d). The upper
slopes of the region are covered with high dissected structural hills with dense pristine forest
(Fig. 2a). The slope ranges between 0 to 44.9 degrees, with lower slopes in the coastal region
mostly inhibited and undergoing rapid coastline modification and Land Use Change. The
North, Northeast, and Southern portions of South Andaman have the steepest slope and relief
area, while the Eastern, Southeastern, and western parts have relatively lower slopes (Fig. 2b
and c). The island has a rough coastline with various bays, inlets, and headlands (Fig. 2). The
Younger coastal plain is a relatively flat and low-lying area adjacent to the coastline, which is
formed through the accumulation of sediments brought by the ocean (Fig. 2e). A wave-cut
platform, formed by the erosive action of waves, are flat or gently sloping rock surface are
found along South Point coastlines in Port Blair (Fig. 2f). These platforms can be exposed at
low tide, which gradually wear away the rock over time, are unique feature of rocky coastlines.
Coral reefs along the coast contribute to the formation of sandy beaches and barrier islands
(Reguero et al., 2018). Mangrove forests are found on coasts in South Andaman Island,
primarily in the salty water and muddy sediments lagoons and tidal zone (Fig. 2g). Mangroves
are crucial in stabilizing coastal ecosystems and providing habitat for various species.
Wandoor, Chidya tapu, and Sippighat are some notable locations of mangrove forests in South
Andaman coastal areas. The coastal plains in south Andaman are dynamic and prone to
tsunamis due to their location and active plate boundary. Therefore, studying shoreline change
and LULC change is especially important because of the potential impacts on local
communities and ecosystems.

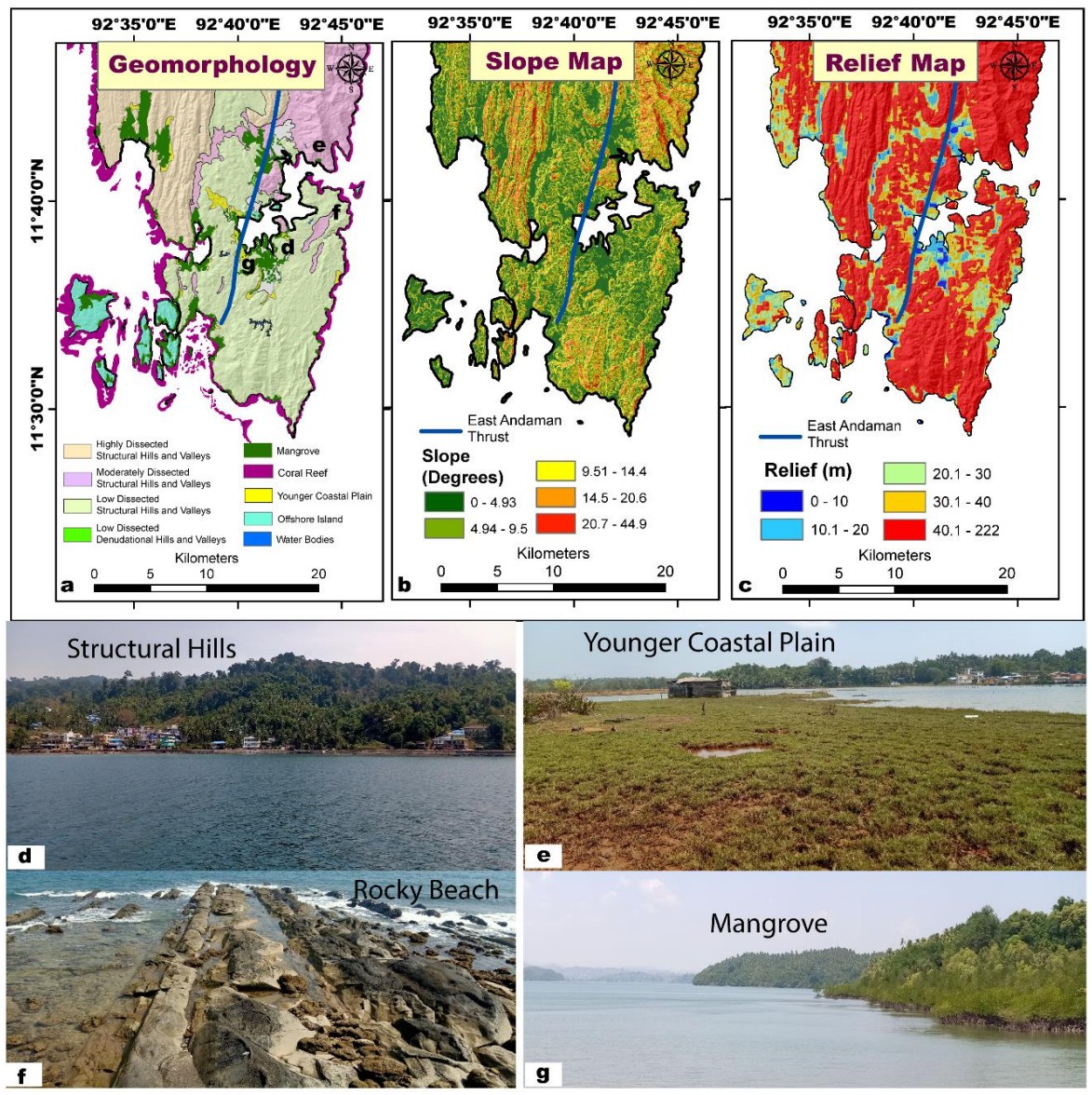

**Figure 2 (a)** *Geomorphology, **(b)** Slope map, **(c)** Relief Map, **(d)** Structural Hills, **(e)** the younger coastal plain, **(f)** Rocky Beach with a wave-cut platform near south point, Port Blair, **(g)** Mangrove.*

## 3. Materials and Methods

It is imperative to generate a spatial dataset that may have a bearing on the dynamic changes to assess the vulnerability.

### 3.1 Data Used

Landsat satellite data, such as Thematic Mapper (TM) and Operational Land Imagery (OLI) sensor for the years 2004, 2005, and 2022, is used to analyze shoreline and monitor the LULC changes along the South Andaman coast in the present study. The Shuttle Radar

Topography Mission (SRTM) Digital Elevation Model (DEM) is used to prepare the study

area's slope and relief map. We used the General Bathymetry Chart of the Ocean (GEBCO) for

run-up and inundation studies along the south Andaman coastal areas (Table 1).

Table 1: Data used in the present study region

| Data | Purpose | Date & Year | Resolution | Sources |
|---|---|---|---|---|
| GEBCO bathymetry | Inundation and Run-Up | 2022 | 90 m | GEBCO (https://www.gebco.net/) |
| Landsat 5 TM, Landsat 8 OLI | LULC and Shoreline Change Analysis | 26-02-2004 27-01-2005 27-02-2022 | 30 m | USGS Earth Explorer |
| SRTM DEM | Slope, Relief | - | 30m | USGS Earth Explorer |
| Geomorphology | Geomorphology | - | 1:250k | bhukosh.gsi.gov.in |
| Socioeconomic data | Population, Tourism, Gross State Domestic Product (GSDP) | 1991-2021 2001-2020 | - | (censusindia.gov.in) (Directorate of economics and statistics) (Rbi.org.in) |

## 3.2 Tsunami Modeling

Several attempts have been made to model tsunamis to calculate inundation and determine run-

up heights to evaluate their impact and hazards along mainland Indian coastal areas and

elsewhere (Cho et al. 2008; Srivastava et al., 2021; Sugawara, 2021; Dani et al. 2023).

## 3.2.1 Tsunamigenic source

Mansinha and Smyile (1971) and Okada (1985) derived closed-form expressions for

the stress and strain field at the source location for different source mechanisms. The focal

mechanism and fault parameters like strike angle, dip angle, slip, and focal depth are necessary

to compute the initial deformation at the source at t=0 sec (Ioualalen et al., (2007), Rani et al.

(2011), Mishra et al. (2014), and Srivastava et al. (2021)). The December 26, 2004, Sumatra

earthquake of magnitude 9.3 had ruptured almost 1400 km. The region is known to have

ruptured into five segments with different slip distributions. Other great tsunamigenic

earthquakes in the Andaman region are the 1881-Car Nicobar and the 26 June, 1941-North
Andaman earthquakes (Table 2).
Table 2: Tsunamigenic earthquake deformation parameters used to simulate different scenarios
a) 1881-Car Nicobar, and b) 1941-North Andaman earthquakes (Mishra et al., 2014), and c)
2004-Sumatra (Ioualalen et al., 2007).

| | 1881-Car Nicobar | 1941 -North Andaman | 2004 Sumatra Earthquake | | | | |
|---|---|---|---|---|---|---|---|
| Input Parameters | | | Seg1 | Seg2 | Seg3 | Seg 4 | Seg5 |
| Longitude (DD) | 92.43 | 92.5 | 94.57 | 93.90 | 93.21 | 92.60 | 92.87 |
| Latitude (DD) | 8.52 | 12.1 | 3.83 | 5.22 | 7.41 | 9.70 | 11.70 |
| Focal Depth (km) | 15 | 30 | 25 | 25 | 25 | 25 | 25 |
| Strike angle (°) | 350 | 20 | 323 | 348 | 338 | 356 | 10 |
| Rake (°) | 90 | 90 | 90 | 90 | 90 | 90 | 90 |
| Slip (m) | 5 | 5 | 18 | 23 | 12 | 12 | 12 |
| Fault Length (km) | 200 | 200 | 220 | 150 | 390 | 150 | 350 |
| Fault Width (km) | 80 | 80 | 130 | 130 | 125 | 95 | 95 |
| Dip (°) | 25 | 20 | 12 | 12 | 12 | 12 | 12 |
| Magnitude (Mw) | 7.9 | 7.7 | 9.3 | | | | |


## 3.2.2 Tsunami wave propagation

The Tohoku University's Numerical Analysis Model for the Investigation of Near field
tsunamis (TUNAMI-N2) to simulate the tsunami run-ups and impact using explicit leap-frog
finite-difference methods by solving nonlinear shallow water wave equations, incorporating
bathymetry, earthquake source parameters, and fault geometry (Imamura and Imteaz, 1995;
Imamura, 1996; Goto, 1997; Imamura et al., 2006; Yalciner et al., 2005). The 2-dimensional
governing equations for tsunami modeling are:
$$\frac{\partial \eta}{\partial t} + \frac{\partial M}{\partial x} + \frac{\partial N}{\partial y} = 0$$

$$\frac{\partial M}{\partial t} + \frac{\partial}{\partial x}\left(\frac{M^2}{D}\right) + \frac{\partial}{\partial y}\left(\frac{MN}{D}\right) + gD\frac{\partial \eta}{\partial x} + \frac{gn^2}{D^{7/3}} M \sqrt{M^2 + N^2} = 0$$

$$\frac{\partial N}{\partial t} + \frac{\partial}{\partial x}\left(\frac{MN}{D}\right) + \frac{\partial}{\partial y}\left(\frac{N^2}{D}\right) + gD\frac{\partial \eta}{\partial y} + \frac{gn^2}{D^{7/3}}N\sqrt{M^2 + N^2} = 0 \qquad (1)$$

In the equation-1, $D$ is the total water depth given by $h+\eta$, $\tau_x$, and $\tau_y$ the bottom frictions
in the x- and y- directions, A the horizontal eddy viscosity which is a constant in space, and the
shear stress on a surface wave is neglected. M and N are the discharge fluxes in the x- and y-
directions which are given by
$$M = \int_{-h}^{\eta} u\,dz = u(h+\eta) = uD \qquad\qquad N = \int_{-h}^{\eta} v\,dz = v(h+\eta) = vD \qquad (2)$$
The bottom friction is generally expressed as follows
$$\frac{\tau_x}{\rho} = \frac{1}{2g}\frac{f}{D^2}M\sqrt{(M^2 + N^2)} \qquad\qquad \frac{\tau_y}{\rho} = \frac{1}{2g}\frac{f}{D^2}N\sqrt{(M^2 + N^2)} \qquad (3)$$
The friction coefficient 'f' and Manning's roughness 'n' are related by
$$n = \sqrt{\frac{fD^{1/3}}{2g}} \qquad (4)$$
It is seen that when D is small and f becomes large then n remains almost a constant.
Substituting M, N, and the above values in fundamental equations of TUNAMI N2 are obtained
which are used to solve the wave propagation using the explicit Leap-Frog finite difference
Scheme as Given by Imamura et al. (2006).
**3.2.3 Computational grid**
In deep-sea regions with longer wavelengths, a coarse grid spacing to model linear effects
is sufficient to resolve the wave with minimal error. As the tsunami wave propagates from deep
to shallow waters,  the wavelength shortens and the amplitude increases, it follows a non-linear
pattern of amplitude dispersion, energy dissipation, and bottom friction and requires finer
resolution grids with more node points to accurately capture the wave dynamics and minimize
errors. The grid spacing should follow the Courant-Friedrich-Lewy conditions for checking the
convergence of the numerical code to a certain asymptotic limit using the following
relationship,
$$\Delta x / \Delta t = \sqrt{(2gh_{max})} \qquad \qquad (5)$$
Where $\Delta t$ and $\Delta x$ are temporal and spatial grid sizes, hmax maximum still water depth in the
computational domain, and g is the gravitational acceleration.
To observe the non-linear or near-shore effects of a tsunami a high-resolution
bathymetry and topography is considered. In the present study, we used GEBCO bathymetry
and topography data formatted into four grids of 81, 27, 9, and 3arc seconds resolutions at a
spacing ratio of 1:3 for grids A, B, C, and D, respectively (Fig. S1). In most computations, the
manning coefficient is around 0.025 as it consists of gravel and sand (Masaya et al., 2020);
however, different manning coefficients can be considered for rough bathymetry (Dao and
Tkalich, 2007). A value of 0.01 is considered for smooth bathymetry and stony cobbles, and a
roughness of 0.035 can be considered. The viscosity and roughness have a certain influence on
mild slopes but it is negligible for steep slopes and a dynamic friction coefficient from 0.01 to
0.1 can be considered (Zhang et al., 2024). For the propagation of tsunamis in shallow water,
the horizontal eddy turbulence terms are negligible as compared with the bottom friction (Dao
and Tkalich, 2007) We simulate the tsunami waves using the TUNAMI-N2 code to get the
directivity map, the wave amplitudes (run-up heights), and inundation distance at different
locations in the study region.
**3.3 Shoreline Analysis in DSAS**
The USGS's digital shoreline analysis system (DSAS) version 5.1 (an ArcGIS
extension) estimates shoreline changes. The procedures are executed in 4 steps: shoreline
digitization, baseline generation, transect generation, and computation of the shoreline change
rate (Raj et al., 2020; Natarajan et al., 2021). The digitized shorelines for 2004, 2005, and 2022
years have been added to a personal geodatabase in a single shapefile. The shoreline image
data is added to the attributes as MM/DD/YYYY, and the baseline is in the meter UTM
projected coordinate system. To estimate rates of change, DSAS uses baseline measurements
of a time series of shorelines and a shapefile (Leatherman, 2003). Generating transects involves
initially choosing a predefined set of parameters from the personal geodatabase, including
settings for the baseline and shoreline. Subsequently, we placed these transects perpendicular
to the shoreline, extending 800 m at intervals of 150 m along the entire shoreline, originating
from the baseline. A 50 m smoothing distance was applied using the 'cast transects' tool within
DSAS to ensure a smoother outcome.
The evaluation of uncertainty encompasses natural and anthropogenic forces such as
wind, waves, tides, currents, and human influences, along with the accuracy of measurement
techniques, including digitization, interpretation, and GPS error. The accuracy of shoreline
position and the rates of shoreline change can be influenced by various error sources, such as
the position of the tidal level, image resolution, digitization error, and image registration
(Jayson-Quashigah et al., 2013; Vu et al., 2021, Basheer et al., 2022). Therefore, the shoreline
positional error (Ea) for each transect was calculated using Equation (6):

$$E_s = \pm\sqrt{E_s^2 + E_w^2 + E_d^2 + E_r^2 + E_p^2} \qquad (6)$$
Where Es is the seasonal error due to seasonal shoreline fluctuations, which is $\sim \pm 5$ m in
extreme ocean level (EOL); Ew is the tidal error, Ed is the digitization error, $E_r$ is the
rectification error and $E_p$ is the pixel error (Fletcher et al. 2011; Vu et al., 2021). This
approach assumes that the component errors are normally distributed (Dar & Dar, 2009).
The total uncertainties were used as weights in the shoreline change calculations. The
values were annualized to provide errors ($E_u$) estimation for the shoreline change rate at
any given transect, expressed in Equation (7):
$$E_u = \pm \frac{\sqrt{U_{t1}^2 + U_{t2}^2 + U_{t3}^2 + U_{t4}^2 + U_{tn}^2}}{T} \qquad (7)$$
where $t_1$, $t_2$, and $tn$ are the total shoreline position error for the various years and T is the
years of analysis.

230       The uncertainty in the shoreline analysis is due to the influence of tides on the Landsat

satellite imagery, which is minuscule in the extensive coastline of the study area. We used
monthly tide gauge data from the Permanent Service for Mean Sea Level (PSMSL) database
(https://psmsl.org/data/obtaining/stations/206.php) at Port Blair station for 2003-2004 and
2017-2021. The data for 2004-2005 and 2022 are unavailable. The tide excursion of 383 mm
or 0.383 m (Fig. S2) is estimated from the highest (1100 mm) and lowest (717 mm) tide gauge
measurements recorded between 2017 and 2020. We calculated uncertainty of 7.21m and
7.12m for 2018-2019 and 2019-2020, respectively, and the same is adopted for 2022 owing to
similar ranges (Table S1). The mean slope of the shore areas is 4-12 degrees near 7 zones. (Fig.
S3, Table S2). We used End Point Rate (EPR) and Net Shoreline Movement (NSM) methods
to analyze the shoreline change (Himmelstoss et al., 2021). To quantify uncertainty, a
confidence interval of 90% and a shoreline uncertainty value of 10m were adopted based on
the recommendations of the United States Geological Survey (USGS) under the National
Assessment of Shoreline Change project (Himmelstoss et al., 2021; Den and Oele, 2018 and
Joesidawati, 2016).
**3.3.1 Net Shoreline Movement (NSM)**
NSM is used to determine the net change in the shoreline position over a specific period by
finding the perpendicular distance between the most recent shoreline (in this case, 2022) and
the oldest shoreline (2004) along each transect. The formula for NSM can be expressed as:

$$NSM = \{d_{2022}\text{-}d_{2004}\}m$$

**3.3.2 End Point Rate (EPR)**

EPR quantifies the shoreline change rate over time and is calculated by dividing the Net

Shoreline Movement (NSM) by the time elapsed between the oldest and most recent shoreline

measurements, which indicates the rate of erosion or accretion. It is important to have data

from at least two shoreline dates (Dolan et al., 1991; Crowell et al., 1997). The formula for

EPR can be expressed as follows:

$$EPR = \left\{ \frac{d2022 - d2004}{t2022 - t2004} \right\}$$

**3.4 Land Use Land Cover Analysis (LULC)**

The LULC map uses Landsat 5 TM (2004 and 2005) and Landsat 8 OLI (2022). False Colour

Composite (FCC) satellite images combine near-infrared, red, and green bands to delineate five

classes: Forest, built-up, Cropland, Water bodies, and Inundated areas. (Prabhbir and Kamlesh,

2011). Tone, texture, size, shape, pattern, association, and other visual interpretation techniques

also were used to interpret different land use classes. Maximum likelihood is a supervised

classification method used in this study to detect LULC change. Each pixel in the classified

Landsat images varies over time due to changes in land cover.

**4. Results**

An analysis of the 2004 tsunamigenic earthquake's impact on the South Andaman

region, focusing on tsunami directivity, arrival times, run-up heights, shoreline changes, and

LULC impact, is examined in detail.

**4.1 Tsunami studies along the South Andaman Region**

We have considered three tsunamigenic seismic scenarios, namely, a) the 1881-Car

Nicobar earthquake, b) the 1941-North Andaman earthquake, and c) the 2004 Sumatra

earthquake, and generated the directivity and run-up maps(Fig. 3). The directivity map shows

that most of the energy propagation is in the East-West direction (Fig. 3 a,b,c), and the
shallower waters surrounding the Andaman and Nicobar Islands has significance influence on
the east-west propagation of tsunamis (Singh et al., 2012). The run-up height along the eastern
coast of South Andaman is greater than the western coast (Fig. 3 b', c', d'; Table 3). This
difference is due to the wider continental shelf on the Western coast of the south Andaman
region and shallow water depths. In the case of a higher magnitude of tsunamigenic earthquakes
in the Car Nicobar or the North Andaman region, higher run-ups will be observed along the
locations, which are considered for the present study (Table 3).
The arrival times of tsunamis vary from 21 minutes to 58 minutes across different locations for
these earthquakes, with the 1881-Car Nicobar earthquake generally resulting in the shortest
arrival time (Fig. 3; Table 3). The run-up heights range from 1-13 m at different locations (Fig.
3; Table 3), which are resultant of earthquake magnitude, the source's proximity to observation
locations, and the local coastal topography that also affected inundations. The extent of
inundation, representing the area covered by the tsunami, ranges from 10m to 950m, with a
wide variation across locations and earthquake events. The 2004 Andaman Sumatra earthquake
resulted in higher run-up heights and inundations compared to the 1881 Car Nicobar, and 1941
Andaman earthquakes and caused extensive damage. Hence, we considered the 2004-
Andaman Sumatra earthquake for a detailed analysis of hazard assessment and scenario
analysis. The arrival times (minutes), run-up height (meter), and inundation extent (meter) at
13 different locations along the South Andaman region for the 2004 Sumatra earthquake (Table
3) are considered for further analysis.

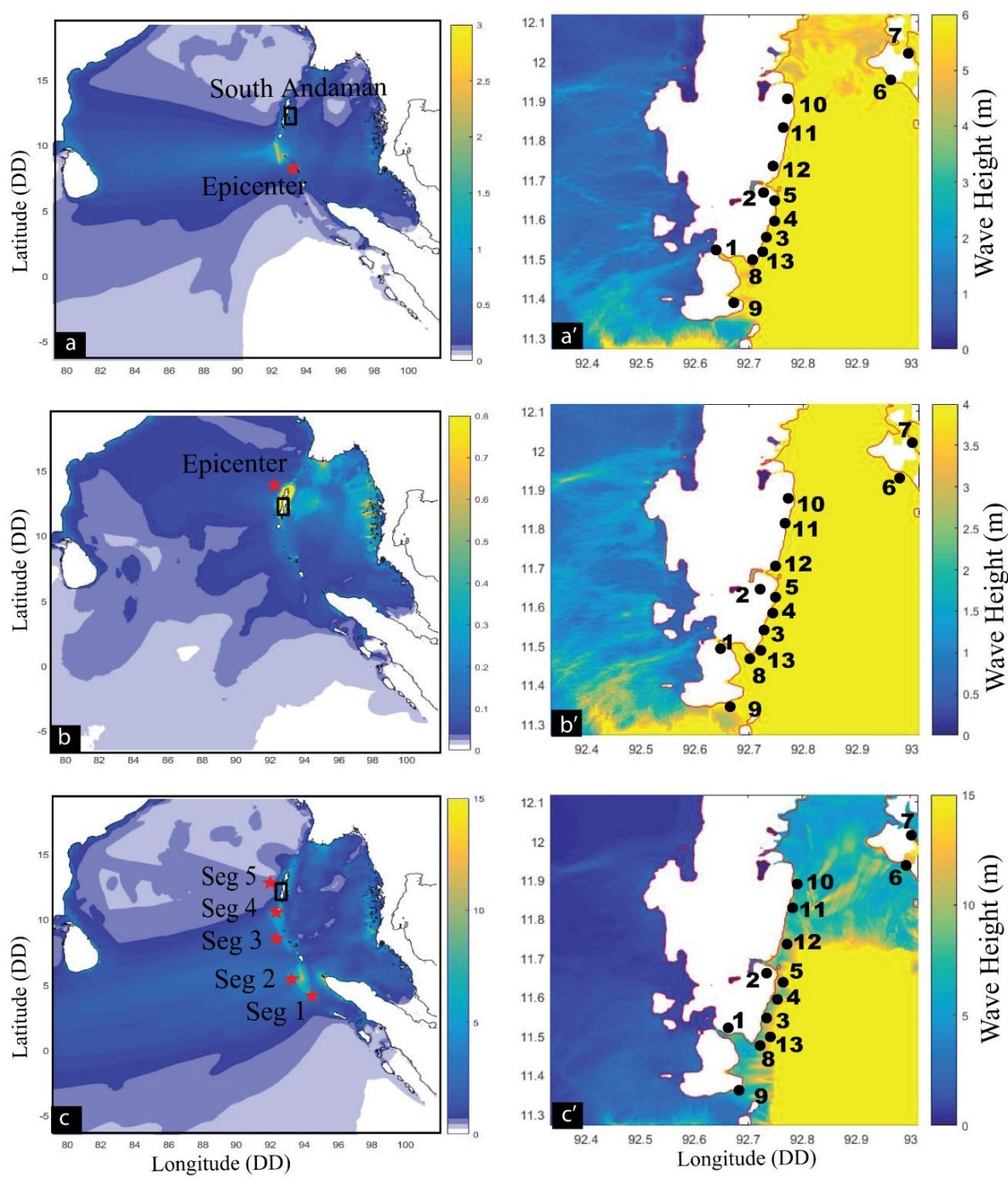


*Figure 3: (a) Directivity and (a') wave run-up height for the 1881-Car Nicobar, (**b** and **b'**) for the 1941-Andaman,*
*and (c and c') for the 2004-Sumatra earthquakes.*

Table 3: Estimated Arrival times, Run-up heights, and inundations at the studied locations from
tsunamigenic a) 1881-Car Nicobar, b)1941-North Andaman earthquakes, and c) 2004-Sumatra
earthquake sources. The SN of locations is common for Figs. 3 and 4.

| SN | Gauge Locations | Longitude Latitude (DD) | Earthquake Sources | Arrival Time (Min.) | Run-up (m) | Inundation (m) |
|---|---|---|---|---|---|---|
| 1 | Wandoorjetty | 92.614750, 11.581667 | a) 1941-North Andaman | 22.5 | 1.25 | 180 |
| | | | b) 1881 Car Nicobar | 32.80 | 2.21 | 200 |
| | | | *c) 2004 - Sumatra* | *36.5* | *3.5* | *450* |
| 2 | Bombooflat | 92.715417, 11.700722 | a) 1941-North Andaman | 24.55 | 2.23 | 350 |
| | | | b) 1881 Car Nicobar | 31.2 | 2.35 | 650 |
| | | | *c) 2004 - Sumatra* | *42* | *5.5* | *90* |
| 3 | Corbyns Cove Beach | 92.770916, 11.642372 | a) 1941-North Andaman | 22.3 | 2.1 | 320 |
| | | | b) 1881 Car Nicobar | 28.8 | 2.3 | 580 |
| | | | *c) 2004 - Sumatra* | *33* | *12.7* | *900* |
| 4 | South Point, Port Blair | 92.702917, 11.652389 | a) 1941-North Andaman | 22 | 2.12 | 280 |
| | | | b) 1881 Car Nicobar | 28.4 | 2.31 | 500 |
| | | | *c) 2004 - Sumatra* | *31.5* | *9.6* | *550* |
| 5 | Thirupatti Temple | 92.703861, 11.581694 | a) 1941-North Andaman | 21.75 | 1.42 | 360 |
| | | | b) 1881 Car Nicobar | 46.5 | 1.65 | 400 |
| | | | *c) 2004 - Sumatra* | *38* | *1* | *200* |
| 6 | Radha Nagar | 92.951722, 11.979306 | a) 1941-North Andaman | 52 | 2.1 | 180 |
| | | | b) 1881 Car Nicobar | 54 | 3.8 | 220 |
| | | | *c) 2004 - Sumatra* | *54* | *2.6* | *156* |
| 7 | Govinda Nagar | 92.989139, 12.030167 | a) 1941-North Andaman | 56 | 1.8 | 220 |
| | | | b) 1881 Car Nicobar | 58 | 3.2 | 190 |
| | | | *c) 2004 - Sumatra* | *58* | *3.6* | *195* |
| 8 | Chidiyatopu | 92.716639, 11.499306 | a) 1941-North Andaman | 21.75 | 1.79 | 300 |
| | | | b) 1881 Car Nicobar | 26.5 | 2.05 | 500 |
| | | | *c) 2004 - Sumatra* | *36* | *3.9* | *585* |
| 9 | Rutland Island | 92.703818, 11.431497 | a) 1941-North Andaman | 25.9 | 1.01 | 585 |
| | | | b) 1881 Car Nicobar | 26.55 | 1.44 | 380 |
| | | | *c) 2004 - Sumatra* | *27* | *6* | *700* |
| 10 | Shoal Bay | 92.795963, 11.934202 | a) 1941-North Andaman | 34.8 | 1.77 | 180 |
| | | | b) 1881 Car Nicobar | 42.5 | 1.45 | 220 |
| | | | *c) 2004 - Sumatra* | *56* | *13* | *950* |
| 11 | Potatang | 92.801282, 12.027380 | a) 1941-North Andaman | 36 | 1.5 | 200 |
| | | | b) 1881 Car Nicobar | 46 | 1.4 | 180 |
| | | | *c) 2004 - Sumatra* | *58* | *12.5* | *210* |
| 12 | Madhuban Bay | 92.785534, 11.782775 | a) 1941-North Andaman | 32 | 1.9 | 180 |
| | | | b) 1881 Car Nicobar | 40 | 1.5 | 200 |
| | | | *c) 2004 - Sumatra* | *54* | *6.9* | *210* |
| 13 | Brichgunj | 92.770162, 11.618980 | a) 1941-North Andaman | 28 | 1.3 | 200 |
| | | | b) 1881 Car Nicobar | 32 | 4 | 300 |
| | | | *c) 2004 - Sumatra* | *30* | *10* | *585* |


Due to the effects of the 2004 tsunami, the stagnation of tsunami water in the
agricultural lands and low-lying areas of the Wandoor region resulted in increased soil salinity
(Fig. 4a); it also damaged the bridge in the Bombooflat area (Fig. 4b), and houses near the
Sippighat area (Fig. 4c, d). Shoal Bay recorded the highest inundation extent of 950m and
experienced the highest run-up height of 13m, indicating significant wave impact (Fig. 3b;
Table 3). Corbyn's Cove Beach and Rutland Island experienced significant inundation
distances exceeding 700m (Fig.3b, Table 3). Potatang, Corbyns Cove Beach, and Brichgunj
also recorded relatively high run-up heights that exceeded 9m (Table 3). Most locations
experienced arrival times between 27 and 58 minutes, indicating a relatively quick propagation
of the tsunami wave. Jain et al. (2005) mentioned that tsunami waves arrived between 40 and
50 minutes in the Andaman and Nicobar Islands. Our results agree with the tsunami run-up
heights estimation by Cho et al. (2008) and Prerna et al. (2015) at a few locations in the present
study area. Since the tide gauge data are available at a few locations along the Indian coast, we
rely on limited field observations along the coast to validate our findings. The field
observations of the water marks on a light post at Bambooflat in Port Blair were seen to be
around 3.8m (Cho et al., 2008), and our computations show it to be ~ 3.5m, within ~7% error
limit. Similarly, at South Point, Port Blair, the field observations are 10m, and our computations
value is 9.6m, which is ~4% deviation, and the deviation is 7% at Chidiyatopu. The Bambooflat
region and Harbour area of Port Blair experienced liquefaction affecting several buildings
(Murty et al., 2006), and our calculations show that the tsunami wave heights were around
5.5m. At most locations, the computed values are within 10% error.
South Andaman experienced significant inundations during the 2004 Sumatra
earthquake, highlighting the urgent need for robust mitigation and preparedness measures in
these vulnerable coastal regions. We aim to contribute to this broader goal by providing
essential data and insights to support evidence-based decision-making and mitigate the adverse

impacts of tsunamis on coastal populations. The study will provide workable input to the local

risk management strategies involving local communities, optimizing evacuation planning,

enhancing early warning systems, fortifying infrastructure resilience, and adopting a multi-

hazard risk assessment approach (National Research Council, 2011).

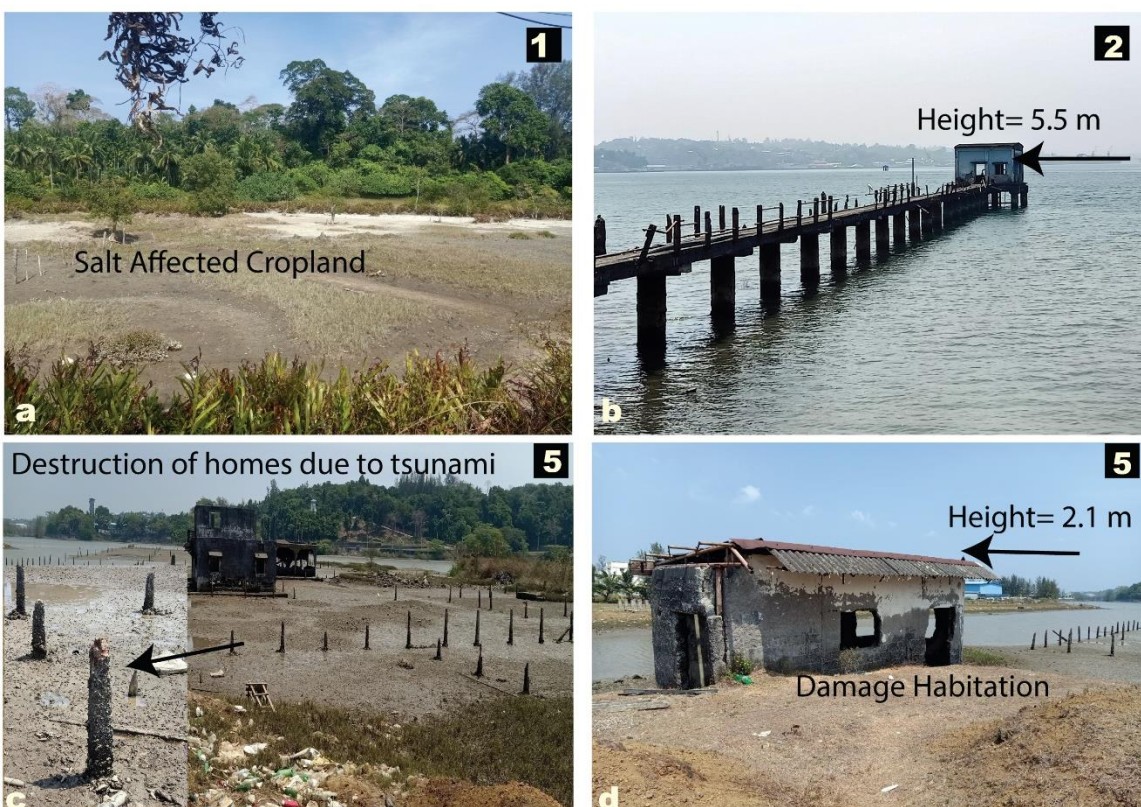

*Figure 4: (a) Stagnation of Tsunami water in the agricultural field and Low-laying areas in Port Blair, (b) damaged bridge in Bombooflat, (c, d) damaged house in the Sippighat area near Port Blair (Photo: 01/03/2023). The number on the field photograph corresponds to respective locations, as in Fig. 3.*

**4.2 Shoreline Change during Tsunami (2004-2005) and post-tsunami (2005-2022)**

The south Andaman coasts are divided into seven zones based on proximity with the

inundation studies to calculate NSM and EPR to understand the short-term and long-term

changes impact of coastal erosion (Fig. 5, Supplement Fig. S4-S10). The NSM and EPR are

calculated over two separate time frames to comprehend the damages caused by tsunamigenic

and regular wind-wave-surge events in South Andaman Island. These zones were used to

understand erosion and accretion rates between (i) 2004 - 2005 (Fig. 5a) and (ii) 2005-2022

(Fig. 5b). The EPR and NSM values from 2004 to 2005 indicate the direct effect of tsunami

waves, whereas 2005 to 2022 values represent periodic wind-wave-surge dynamics. Periodic

coastal shoreline changes refer to the regular and repeating fluctuations in the position of the

shoreline along the coast. Natural and human-induced factors can influence these changes. A

total of 1,083 transects are created at 50-m intervals, distributed among the zones as follows:

Zone 1 (339 transects), Zone 2 (147 transects), Zone 3 (89 transects), Zone 4 (74 transects),

Zone 5 (137 transects), Zone 6 (73 transects), and Zone 7 (220 transects). The shoreline

variation rates indicate positive accretion and negative erosion (Fig. 6, Table 4). The EPR

Changes in meters per year (m/y) for the periods 2004-2005 show a higher erosion rate

compared to 2005-2022, particularly in Zones 3, 4, and 5 (Fig. 6a). The NSM focused on two

distinct time frames, indicate the NSM rates during the tsunami, for the year of 2004-2005, and

the NSM rates over the extended 17-year period from 2005 to 2022 are measured in meters

(Fig. 6b). The detailed analysis of the maximum (accretion), minimum (erosion), and mean

shoreline changes for each of the seven zones that occurred during the tsunami event and the

post-tsunami period are discussed below.

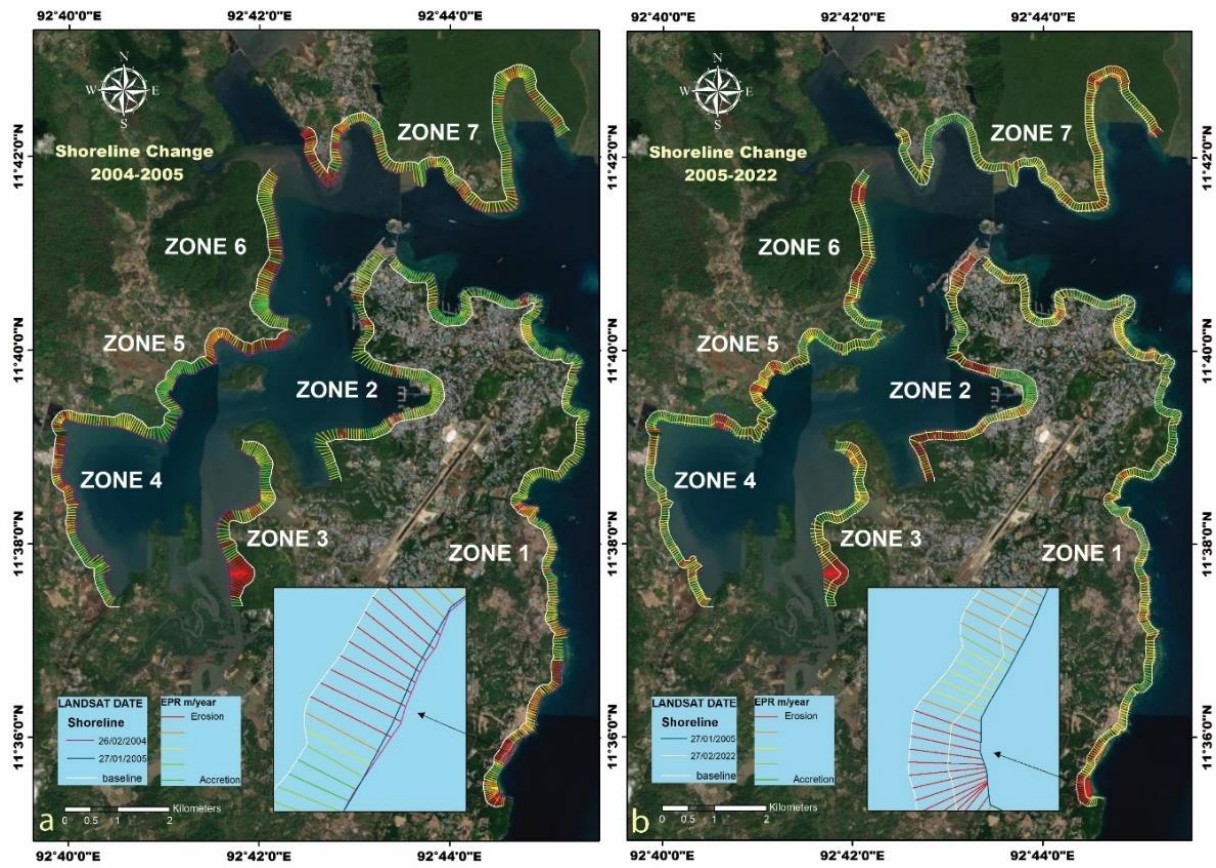

358

*Figure 5: Shoreline changes observed (a) during 2004-05 due to the tsunamigenic process and (b) from 2005-2022 due to wind wave surges overlaid on Google Earth images (@Google Earth). The affected coastline is subdivided into seven distinct zones for detailed analysis.*

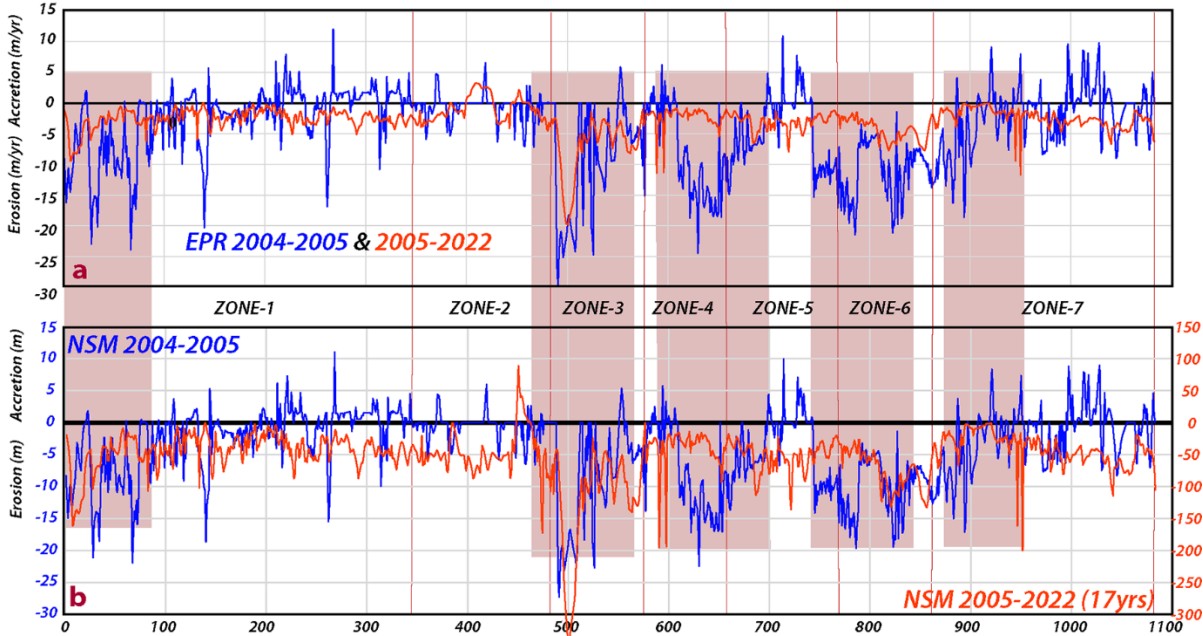

362

*Figure 6: (a) The rates of erosion and accretion in seven distinct Zones along the South Andaman shoreline using EPR methods, and (b) NSM have been conducted between the years 2004-2005 and 2005-2022. Highlighted color indicating high erosion zone*

366

Table 4 Shoreline change in southern Andaman is observed for 2004-2005 and 2005-2022 using USGS's DSAS methods (Himmelstoss et al., 2021).

| ZONE | | 2004-2005 | | 2005-2022 | |
|---|---|---|---|---|---|
| | | EPR(m/y) | NSM(m) | EPR(m/y) | NSM (m) |
| ZONE 1 | Mean | -2.85 | -2.62 | -2.55 | -43.57 |
| | Minimum | -23.9 | -21.29 | -9.44 | -161.21 |
| | Maximum | 12.05 | 11.06 | 0 | 0 |
| ZONE 2 | Mean | -0.54 | -0.50 | -1.0639 | -18.174 |
| | Minimum | -7.17 | -6.58 | -4.56 | -77.93 |
| | Maximum | 6.54 | 6 | 3.25 | 55.56 |
| ZONE 3 | Mean | -9.92 | -8.11 | -7.10 | -121.51 |
| | Minimum | -24.71 | -23.27 | -19.87 | -339.51 |
| | Maximum | 5.58 | 4.37 | -1.02 | -17.42 |
| ZONE 4 | Mean | -7.92 | -7.72 | -2.24 | -38.34 |
| | Minimum | -24.47 | -22.46 | -11.42 | -195.03 |
| | Maximum | 6.23 | 5.72 | -0.79 | -13.42 |
| ZONE 5 | Mean | -6.594 | -6.05 | -2.94 | -50.26 |
| | Minimum | -21.47 | -19.7 | -7.95 | -135.83 |
| | Maximum | 10.88 | 9.99 | -1.03 | -17.54 |
| ZONE 6 | Mean | -9.74 | -8.94 | -4.92 | -84.05 |
| | Minimum | -21.18 | -19.44 | -7.75 | -132.39 |
| | Maximum | -1.46 | -1.34 | -1.86 | -31.73 |
| ZONE 7 | Mean | -2.16 | -1.986 | -2.43 | -41.56 |
| | Minimum | -18.65 | -17.29 | -11.7 | -199.96 |
| | Maximum | 9.77 | 8.97 | -0.04 | -0.61 |

ZONE 1: This zone experienced a combination of erosion and accretion between 2004-05 and 2005-21. The maximum erosion rates are observed at Megapoda, with an EPR of -23.9 m/y. and -9.44 m/y., NSM analysis shows the estimated erosion is -21.29m and -161.21m respectively (Fig. S4 a, b, Table 4). The southern part of South Andaman Island has more shoreline erosion rather than accretion, which can be attributed to the heightened impact of tsunamis on the southern region, a phenomenon that is more significant when compared to the northern part of South Andaman Island. These Sediments eroded from one coastline area are often transported along the shoreline by the longshore currents. The angle of wave approach creates these currents and is responsible for moving sediment parallel to the coastline.

ZONE 2: This zone experienced a combination of erosion and accretion between 2004-05 and 2005-21. The maximum rate of erosion is -7.17 m/y and -4.56 m/y (EPR) was recorded

at IOC Colony, while the maximum accretion rate of 6.54 m/y and 3.25 m/y (EPR) was
observed at Ashwin Nagar Respectively. The NSM analysis indicated a shoreline retreat
of -6.58 m at IOC Colony and -77.93 m advancement at Ashwin Nagar. The jetties in the
Jungli Ghat port played a role in controlling erosion and accretion at these sites (Fig. S5,
Table 4).
ZONE 3: This zone experienced a combination of erosion and accretion between 2004-05 and
2005-21. The maximum erosion rate is -24.71 m/y and -19.87 (EPR) at Flat Bay, while
the maximum accretion rate is 5.58 m/y and (EPR) at NLC Limited. The NSM analysis
revealed a shoreline retreat of -23.27 m and -339.51 m at Flat Bey. High wave energy
and exposure to strong currents, which are more common near Flat Bay, can lead to
increased erosion of mangrove shorelines (Fig. S6, Table 4).
ZONE 4: This zone experienced a combination of erosion and accretion between 2004-05 and
2005-21. The maximum erosion rate is -24.47 m/y at Ferrargunj and -11.24 m/y (EPR)
at PLK Creek Resort, NSM estimated erosion is -22.46 m and -195.03m at Chouldari
(Fig. S7). We observed the shoreline erosion area using the Landsat time-lapse satellite
images between 2004-2005, and 2022 near Flat Bay, South Andaman, has revealed
noteworthy environmental changes. The dark blue color observed in 2004 and 2005
indicates the presence of deep-water bodies, whereas the light blue color in the 2022
image suggests the water bodies have become shallow with significant fresh sediment
load (Fig. 7; Table 4).

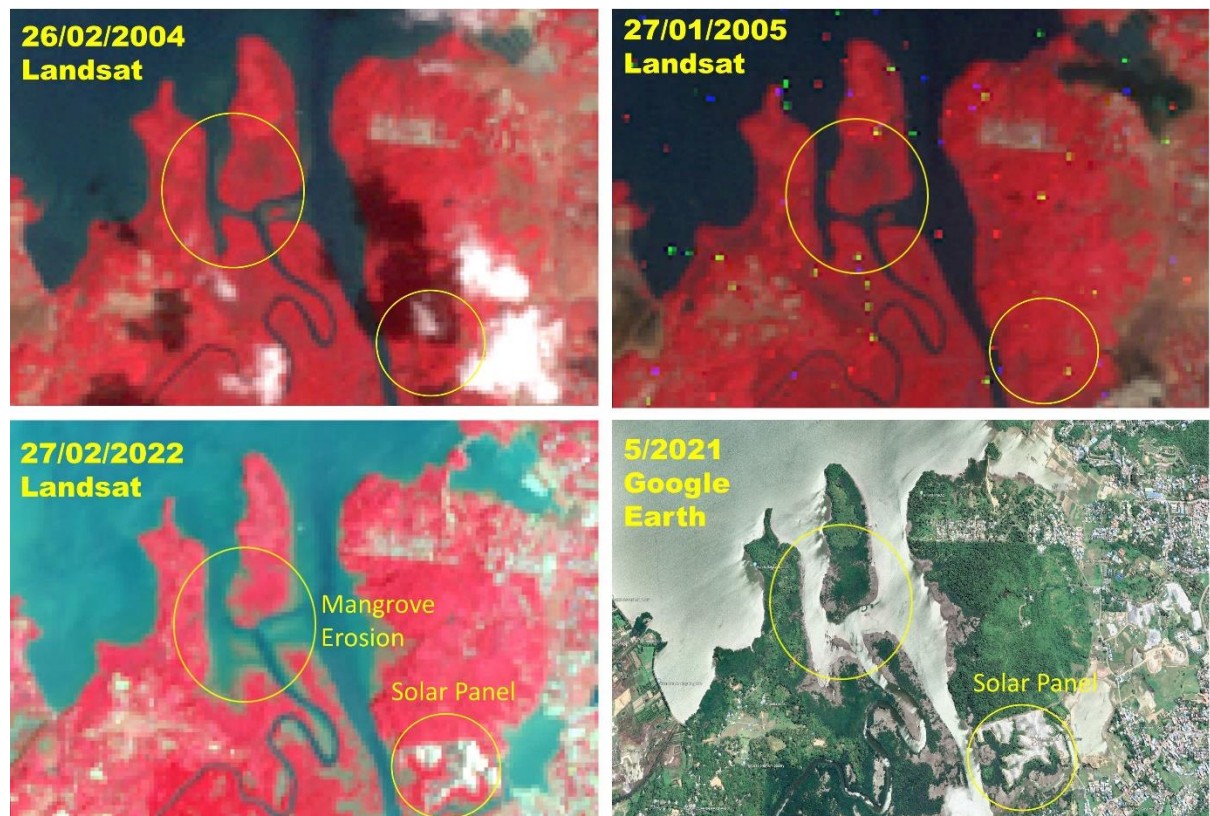


*Figure 7: shows a time-lapse satellite imagery of Landsat 8 FCC near the Flat Bay area (marked in yellow circle)*
*during the years 2004 and 2005 showing robust mangrove coverage is evident. However, when comparing the*
*Landsat 8 image in 2022 and the corresponding Google Earth image (@Google Earth), it is apparent that the*
*mangrove ecosystem in this area has experienced substantial erosion and the development of Solar panels.*

ZONE 5: The maximum erosion rate of -21.47 m/y (2004-05) and -7.95 (EPR 2005-22) is
recorded at Mithakhari. According to the NSM analysis, the shoreline retreated by -19.7
m and -132.39m at Mithakhari (Fig. S8). In this zone, Coastal development,
infrastructure construction, and alteration of natural hydrological patterns can disrupt
sediment transport and exacerbate erosion (Fig. 8; Table 4).

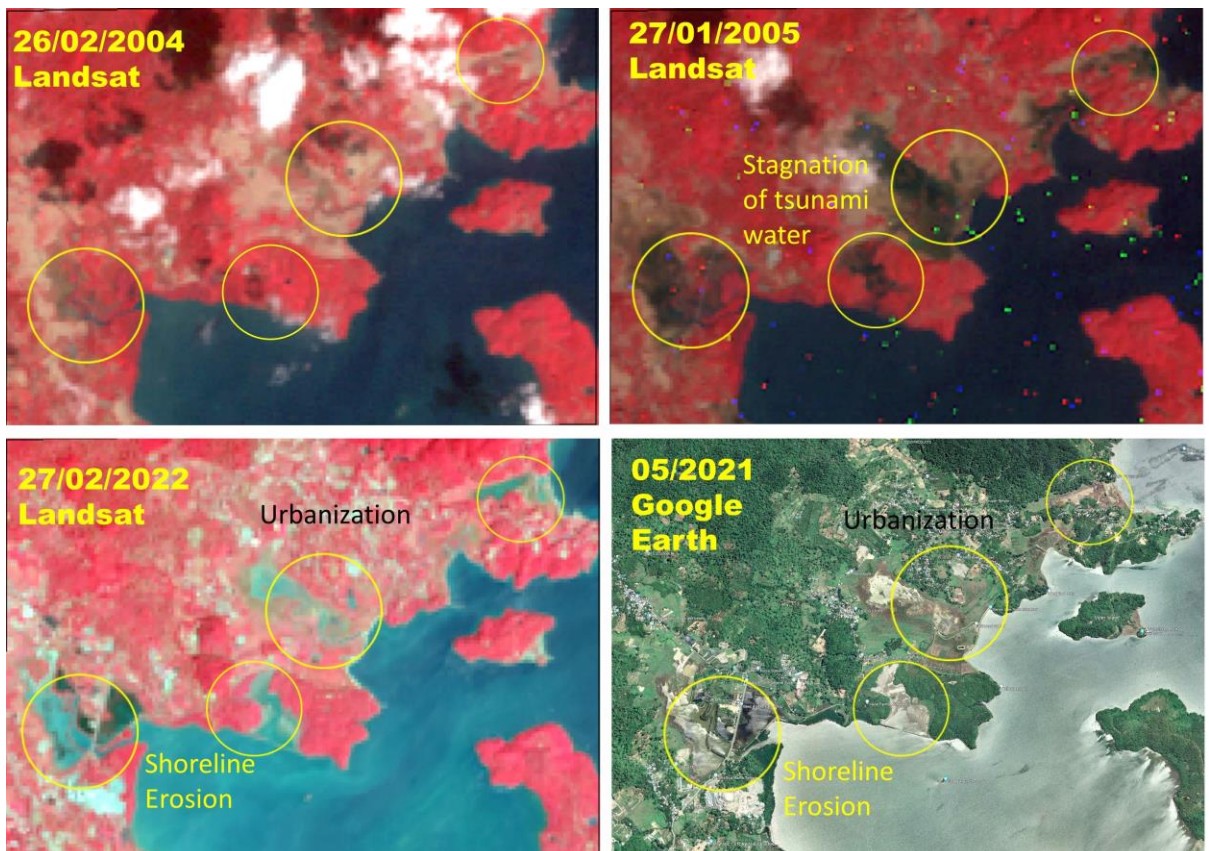

*Figure 8: shows Landsat 8 time-lapse imagery and © Google Earth imagery near the Ograbraj and Mithakhari region depicting the erosion activity during and after the tsunami and the imagery shows a significant growth in the built-up areas surrounding the tsunami-affected areas in 2004.*

ZONE 6: This zone is predominantly affected by erosion, with no observed accretion. The maximum erosion rate is -21.18 m/y and -7.75 m/y (EPR) at Namunaghar, and the NSM estimated erosion is -19.44 m and -132.39m at Namunaghar (Fig. S9). In February 2004, immediately before the catastrophic tsunami event, there was no observable presence of stagnant water in the area (Fig. 9). However, by January 2005, following the tsunami, the images distinctly exhibited the stagnant water. In February 2022, the same location exhibited substantial shoreline erosion within the extensive mangrove and agricultural area, accompanied by increased urban development along the shoreline. The progression of urban development was also validated using Google satellite imagery. The sediment carried by ocean currents deposited in low-lying areas revealed caused shallowing and significant changes in ocean water color.

431

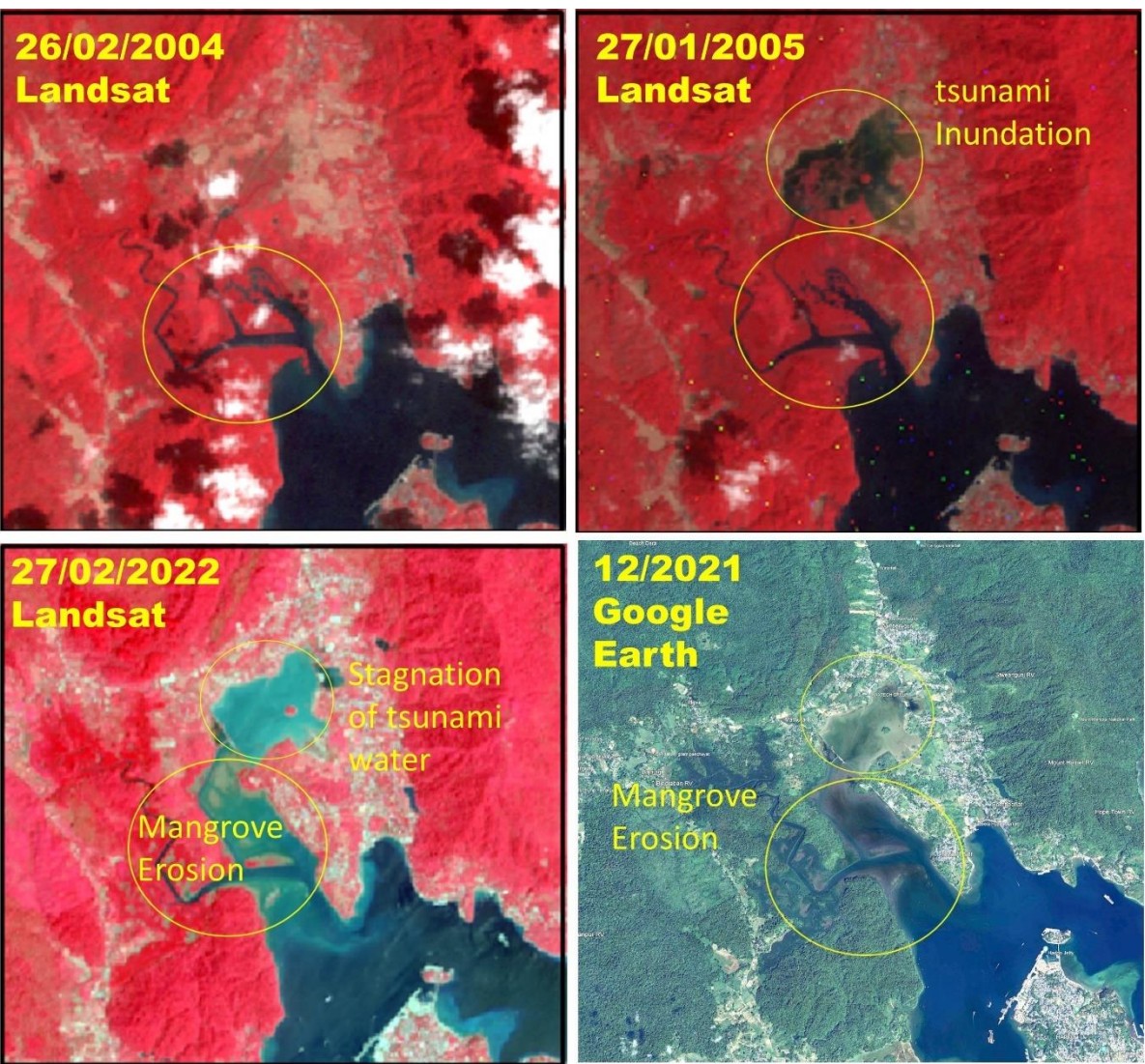

432

*Figure 9: shows the Change detection of the shoreline using Landsat 8 time-lapse imagery and © Google Earth*
*imagery for 2004 before, 2005 after the tsunami, and the 2022 present status of the shoreline.*

ZONE 7: This zone experienced a combination of erosion and accretion between 2004-05 and

2005-21. The maximum erosion rate is -8.36 m/y and -11.7 m/y (EPR) at Shore Point,

while the maximum accretion rate is 9.77 m/y (EPR). The NSM analysis indicated an

erosion of -17.29 m at Shore Point and -199.96 m at North Bay (Fig. S10; Table 4).

Notably, a tsunami with a height of 9.6 m was observed at Shore Point.

The natural rate of shoreline movement in the South Andaman region has increased

following the tsunami event, which is attributed to several factors, including the removal of

vegetation cover, the softening of exposed bedrock, and the destabilization of unconsolidated
materials caused by the tsunami, all of which have made the region more susceptible to erosion
(Yunus et al., 2016). Comparing the erosion and accretion rates suggests the erosion rates were
significantly less during the 2005-2022 period in comparison to the 2004-05 tsunami,
highlighting the adverse effect of the tsunami.
**4.3 Land Use and Land Cover (LULC) Analysis**

449         The LULC is categorized into 5 distinct classes: Built-up, Forest, Inundation, Cropland,

and water Bodies (Fig. 10). The overall accuracy obtained is 90.11%, 89.96%, and 90.30%
with a quantitative assessment of $K_{hat}$ (Kappa) coefficient is 0.78, 0.762 and 0.79 for 2004,2005
and 2022 images, respectively (**Table S3**). Our primary objective is to determine the extent of
land use pattern changes from 2004 to 2022 in areas affected by the 2004 tsunami. Several
researchers have already examined the vulnerability and impact of the 2004 tsunami on South
Andaman, including (Velmurugan et al., 2006; Sachithanandam,2014).

456         The LULC classification for the South Andaman region in tsunami-impacted areas in

the years 2004, 2005, and 2022 reveals significant changes (Fig. 10, Table 5). 1) The built-up
area decreased from ~7.38% in 2004 to 6.23% in 2005, marking a 1.15% decrease. However,
it subsequently increased by 11.11% by 2022. 2) Cropland coverage decreased from around
22.12% in 2004 to ~11.93% in 2005, indicating a substantial reduction of 10.19%. It then
increased to 17.15% by 2022. 3) Inundation areas increased from about 3.29% in 2004 to
27.65% in 2005, showing a notable rise of 24.36%. However, by 2022, they decreased by
~18.57%. 4) Forested areas saw a significant decrease from ~66.46% in 2004 to about 51.10%
in 2005, signifying a reduction of 15.36%. This decrease persisted in 2022, remaining at
~51.10%. 5) Water bodies covered around 0.62% of the area in 2004, which increased slightly
to about 0.76% in 2005. By 2022, there is a more significant increase, reaching 2.05%.

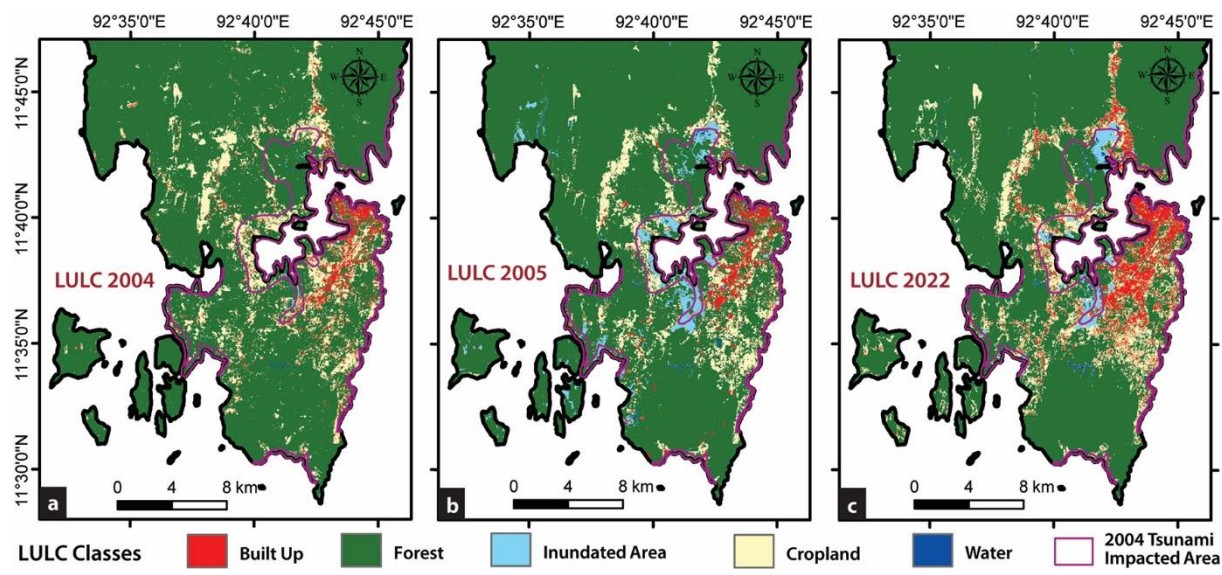


*Figure 10: (a) LULC 2004 (b) LULC 2005, and (c) LULC 2022 in tsunami-impacted areas (pink color) and South*
*Andaman.*
Table 5: LULC Analysis for 2004, 2005 to 2022 in tsunami impacted area

| LULC classes | 2004 Area in km$^2$ | 2004 % of Area | 2005 Area in km$^2$ | 2005 % of Area | 2022 Area in km$^2$ | 2022 % of area |
|---|---|---|---|---|---|---|
| Built-Up | 3.57 | 7.38 | 3.01 | 6.23 | 5.38 | 11.11 |
| Forest | 32.19 | 66.46 | 25.79 | 53.40 | 24.74 | 51.10 |
| Inundation Area | 1.64 | 3.39 | 13.36 | 27.65 | 8.99 | 18.57 |
| Cropland | 10.71 | 22.12 | 5.76 | 11.93 | 24.74 | 17.15 |
| Water Bodies | 0.30 | 0.62 | 0.36 | 0.76 | 0.99 | 2.05 |
| Total Area (Sq. Km) | 48 | 100 | 48 | 100 | 48 | 100 |


The LULC classification for the South Andaman region in the years 2004, 2005, and 2022
shows significant changes (Figure 10, Table 6)
**1) Built-Up Area**: In 2004, the built-up area covered 19.92 km², constituting ~3.84% of the
total study area. By 2005, this area had reduced to 17.66 km², accounting for 3.41% of
the total area. by 2022, there was a significant expansion, with the built-up area
occupying 45.07 km², representing 8.68% of the total region.
**2) Forest**: In 2004, forests dominated the landscape, covering 432.85 km², which was
approximately 83.43% of the total study area. By 2005, this forested area slightly
decreased to 420.79 km², comprising 81.27% of the total area. However, by 2022, the
forest cover continued to decline, with an area of 408.66 km², accounting for 78.78% of
the total region.
**3) Inundation Area**: In 2004, the inundation area was limited, covering 3.40 km² or 0.65% of
the total area. In 2005, there was a substantial increase, expanding to 28.41 km², which
represented 5.48% of the total area. By 2022, the inundation area decreased to 13.89 km²,
making up 2.66% of the total region.
**4) Cropland**: Cropland covered 61.77 km² in 2004, accounting for 11.90% of the total study
area. By 2005, this area reduced to 49.34 km², representing 9.53% of the total area. In
2022, the cropland area further decreased to 48.65 km², making up 9.37% of the total
region.
**5) Water Bodies**: In 2004, water bodies covered a small area of 0.83 km², approximately 0.16%
of the total area. By 2005, this area slightly increased to 1.54 km², constituting 0.29% of
the total region. There was a more significant expansion during 2022, with water bodies
occupying 2.45 km², accounting for 0.47% of the total area.
Table 6: LULC Analysis for 2004, 2005 to 2022 in the Study region

| LULC | 2004 Area in km$^2$ | 2004 % of Area | 2005 Area in km$^2$ | 2005 % of Area | 2022 Area in km$^2$ | 2022 % of area |
|---|---|---|---|---|---|---|
| Built-Up | 19.92 | 3.84 | 17.66 | 3.41 | 45.07 | 8.68 |
| Forest | 432.85 | 83.43 | 420.79 | 81.27 | 408.66 | 78.78 |
| Inundation Area | 3.40 | 0.65 | 28.41 | 5.48 | 13.89 | 2.66 |
| Cropland | 61.77 | 11.90 | 49.34 | 9.53 | 48.65 | 9.37 |
| Water Bodies | 0.83 | 0.16 | 1.54 | 0.29 | 2.45 | 0.47 |
| Total Area (Sq. Km) | 518 | 100 | 518 | 100 | 518 | 100 |





**5. Discussion**

The complex interaction between geomorphology, shoreline change, LULC changes, and economic factors in tsunami vulnerability and impact assessment in South Andaman is discussed below;

**5.1  Shoreline changes VS LULC**

The impact of tsunamis varies due to differences in landforms, relief, slope, elevation, and the presence (or absence) of natural barriers such as coral reefs and mangroves. It has been observed that for a given water depth on the shelf, if the continental slope is steeper, greater mangrove cover, greater relief, and higher elevation can result in a greater amount of energy being reflected, leading to a smaller tsunami wave height on the shelf. On the other hand, with a flatter slope, low relief, and less vegetation cover area on the coastal side, the reduced reflection and effect of shoaling can increase tsunami wave height (Siva et al., 2016). Coastal erosion is a natural process in south Andaman that occurs when waves, currents, tsunamis, and tides erode the shoreline, removing sediment and land over time. Factors such as sea-level rise, wave energy, storm events, and human activities can contribute to increased rates of erosion.

Over time, the geomorphological landforms continue to shape and modify the landscape. However, human activities and developmental pressures are significant drivers of LULC change in South Andaman (Fig. 10 a, b, c). Common LULC changes observed in the area include deforestation for urban expansion, conversion of land for agriculture, infrastructure development, and alterations to the coastal zone (Yuvaraj et al., 2014; Thakur et al., 2017; Jaman et al., 2022). The interaction between geomorphology and LULC change is particularly evident in the coastal regions of South Andaman, where coastal erosion and accretion processes influence both LULC patterns and development decisions. The erosion occurring near the shoreline leads to the loss of valuable land, affecting agricultural areas and forest regions (Fig. 7,8,9). Conversely, accretion processes can contribute to the growth of coastal areas by building

new landforms and influencing land use decisions in those locations (Nagabhatla et al., 2006;
Ali and Narayana, 2015; Mageswaran et al., 2021).
**5.2  Inundation and run observation**
Our computations have shown that the tsunami wave heights for around 5.5 m inundation 90
m are observed in Bombooflat (Fig.4b). Similarly, the harbor area of Port Blair has seen
structural failures in some building's foundations, and our computations show wave heights of
3.6m in that area. Chidiya Tapu, which is 25 km from Port Blair, the estimated run-up is 3.9
m, and the inundation is 585 m, which shows a gradual slope in the region (Fig. 2). Coming to
the Southpoint Magar area (Port Blair), a high run-up of 8.5 m is computed, and the inundation
level is 550 m. Houses located near the open sea were completely washed away. At Wandoor
Jetty in Port Blair, the calculated run-up is 3.46, the inundation is 450m, and the saltwater
intrusion was observed due to the tsunami.
**5.3  LULC vs economic change:**

538       The presence of people, infrastructure, or assets in a hazard-prone location is referred to

as exposure, and vulnerability is the degree to which a person, community, or system is
susceptible to the impacts of a hazard. Vulnerability is determined by physical, social,
economic, and environmental factors. (United Nations Office for Disaster Risk Reduction).
Several factors can contribute to changes in exposure, such as population growth, Industrial
development, and LULC change. It is anticipated that the population of the Andaman and
Nicobar Islands will double by 2050 (Nanda and Haub, 2007), and the islands are experiencing
an increasing influx of tourists. The increased population density in these regions intensifies
the strain on already vulnerable lands. As a result, when a disaster, such as a natural calamity,
occurs in these areas, it affects the tourists and has severe repercussions for the large local
population heavily dependent on tourism-related activities (Annan et al., 2005; Wood et al.,
2019; Sathiparan et al., 2020, Hamuna et al., 2019). The increases in population from 1971 to
2020, as well as built-up areas, are shown before and after the 2004 tsunami, and GSDP from
2001 to 2020 in tsunami-prone areas of South Andaman are observed in Fig. 11.

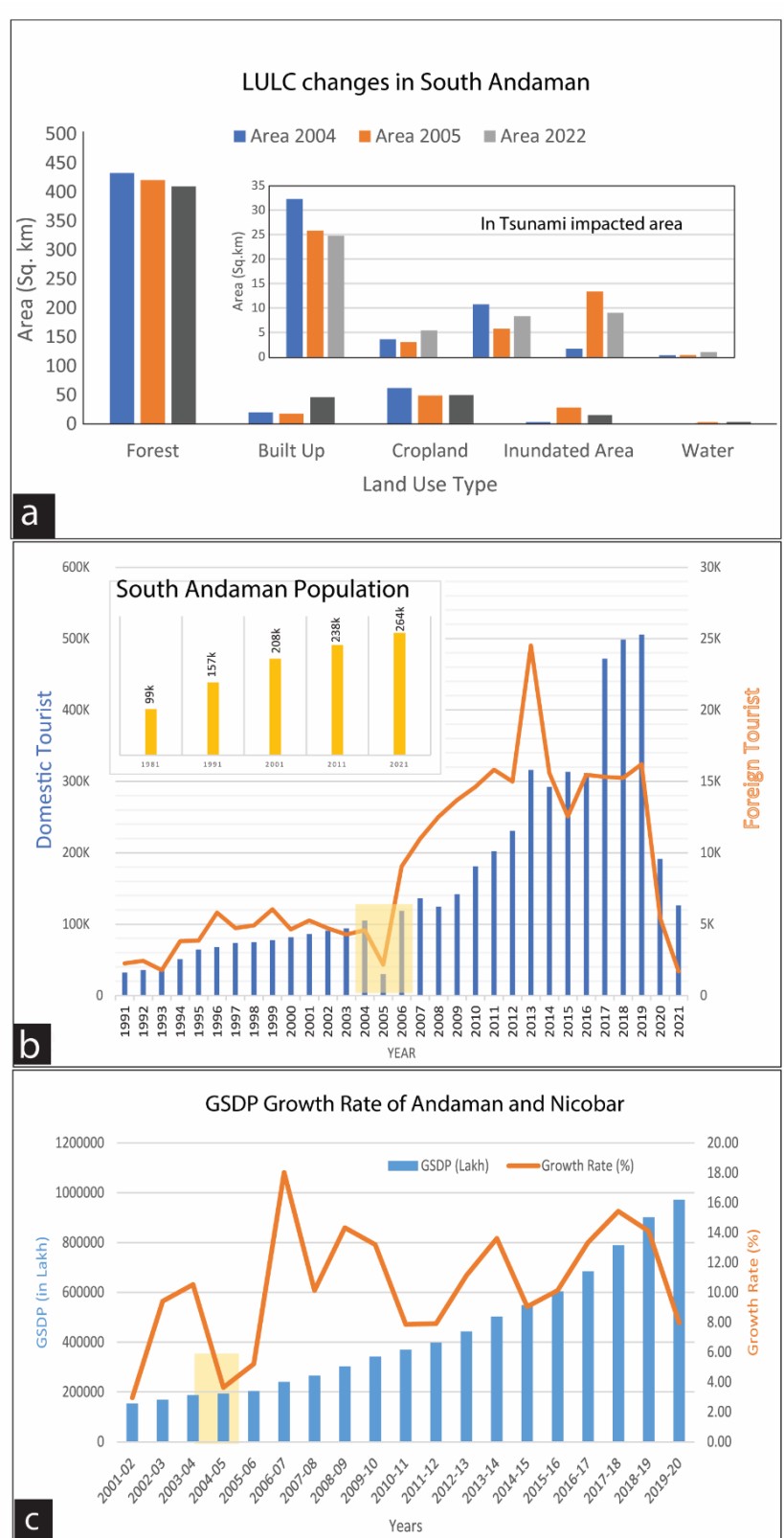

*Figure 11: (a) LULC change in south Andaman and also in tsunami-affected areas of 2004. The LULC classification*
*reveals that there has been a significant increase in built-up areas, inundated areas, and water bodies, while the*

*agricultural land and vegetation have decreased. The increasing trends of tourists and local population in south*
*Andaman can be seen in Fig. (b). The GSDP growth rate shows the macroeconomic impact on GSDP in 2005 due*
*to the tsunami impact (c).*

The increase in built-up areas could also positively impact the GSDP by boosting the
construction and real estate sectors and providing more job opportunities in the tourism and
hospitality industries (Fig. 11a). The 2004 Indian Ocean tsunami significantly impacted the
GSDP of the Andaman and Nicobar Islands, particularly in the tourism and fisheries industries
(Fig 11c). According to a report by the National Institute of Disaster Management, the
Andaman and Nicobar Islands suffered losses amounting to INR 7.5 billion due to the 2004
tsunami, with damages to the tourism industry being the most significant. It is important to
carefully manage this growth and ensure sustainable development practices protecting both the
natural environment and the local population's well-being. This includes implementing
effective disaster preparedness measures, promoting sustainable tourism practices, and
balancing economic development with environmental conservation in the region.
**5.4 Implication for changing scenario of vulnerability**
India Inc. estimates that the total losses surpassed Rs 3,000 crore. Specifically, the losses in
Andaman & and Nicobar Islands exceeded Rs 1,000 crore as per industry estimates
(Economictimes.com). If a tsunami of similar magnitude were to occur again, the economic
loss would be five times as high as those experienced in 2004. After the 2004 tsunami, the
coastal area experienced significant development, with built-up areas expanding in already
affected areas from ~7.38 % in 2004 to ~11.11 % in 2022. This increase in urbanization and
infrastructure means that more properties, businesses, and critical facilities are now located in
the coastal zone. The affected region's local population grew from 208k in 2001 to 264k in
2021 (Figure 11b). With more people living in the coastal area, there is a higher risk of
casualties and a greater demand for resources and aid during and after a tsunami. The number
of tourists visiting the coastal area has increased significantly, from 98,000 tourists in 2001 to
500,000 by 2019 (Figure 11b). Tourists are generally less familiar with local hazards and
evacuation routes, making them more vulnerable during a tsunami. The presence of a large
number of tourists can add complexity to evacuation and relief efforts, potentially leading to
higher economic losses. The region has experienced a sharp decline in forest and cropland
areas. Forests act as natural buffers, helping to reduce the impact of a tsunami by absorbing
some of the wave energy. Additionally, the loss of cropland can disrupt the supply chain during
and after a disaster, affecting food availability and leading to economic losses beyond property
damage.
**6. Conclusions**
The South Andaman region is vulnerable to tsunamis due to its location in the seismically
active zone. In such an environment, tsunami preparedness and resilience are crucial. This
includes implementing effective early warning systems, raising public awareness, and
strengthening infrastructure resilience. Incorporating ecosystem-based approaches, such as
preserving and restoring natural coastal land, can also contribute to reducing tsunami
vulnerability. The South Andaman region is prone to shoreline changes due to natural processes
and human activities. Regular monitoring and assessing these changes is crucial to
understanding their impacts on coastal ecosystems and communities. Implementing
appropriate coastal management strategies, such as beach nourishment, dune restoration, and
erosion control measures, can help mitigate the negative effects of shoreline changes. It is
important to adopt sustainable land use practices that balance economic development with
resource conservation and responsible use. This involves promoting eco-friendly tourism,
protecting sensitive ecosystems like mangroves and coral reefs, and implementing land use
planning that considers the carrying capacity and vulnerability of the region. Tsunami modeling
along the coastal locations shall help decision-makers how to construct structures along the
coast. Decision makers will also be able to quantify the tsunami impact on sloping beaches,
Flat beaches, and areas having boulders/mangroves. Engaging local communities,
stakeholders, and indigenous knowledge holders in decision-making processes and promoting
capacity-building initiatives are critical for ensuring the sustainable development of the
Andaman region.
**Code availability**
No
**Data availability**
All data included in this study are available upon request by contacting the corresponding
author**.**
**Authors' contributions**
Vikas Ghadamode: Computations, Fieldwork, and Manuscript Writing.
K. Kumari Aruna: TUNAMI-N2 Computation and Fieldwork, Manuscript Writing
Anand Kumar Pandey: Manuscript Editing and Contribute Ideas and Suggestions
Kirti Srivastava: Paper Writing and TUNAMI-N2 Computations
**Competing interests** / **Conflicts of interest/**
The authors declare that they have no known conflicts of interest.
**Declarations**
The authors declare that they have no known conflicts of interest.
**Financial support**
No Funding
**Acknowledgements:**
The authors acknowledge encouragement and permission to publish from the Director, CSIR-
NGRI (Ref. No. NGRI/Lib/2024/Pub-51). VG acknowledges UGC, India, for SRF for
pursuing a PhD (Grant no.: 10/UGC-JRF/209/19-ESTT).

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
