# Peer review of "Shoreline and Land Use Land Cover Changes along the 2004 tsunami- affected South Andaman Coast: Understanding Changing Hazard Susceptibility"

_Natural Hazards and Earth System Sciences, 2023_

## Author Comment (AC1)

**Response to Reviewer 1**

**Ms. No: Preprint nhess-2023-191**

**Title**: **Shoreline and Land Use Land Cover Changes along the 2004 tsunami-affected South Andaman Coast: Understanding Changing Hazard Susceptibility. https://doi.org/10.5194/nhess-2023-191**

**In this response file, the text in blue shows the comments from reviewers, while the text in black is our replies.**

**Reviewers comment**

This paper addresses an important tsunami risk problem, which is aggravated by the increasing trend in population exposure, tourism etc. The analysis undertaken identifies a number of important tsunami risk issues, but falls short of what is required for a robust population tsunami safety study. To make the conclusions more robust, some additional scenario analyses would be insightful and instructive. First, all tsunami wave height outcomes are subject to a substantial degree of stochastic variability. Venturing beyond the actual 2004 tsunami wave height measurements, the implications for local upward variations in wave height should be considered by perturbing the tsunami source dynamics. Furthermore, other potentially dangerous earthquake-generated tsunamis merit attention, and an ensemble of some alternative potential tsunami scenarios should be considered, especially those which impact regions of recent economic development. This broadening of the basic tsunami modelling content of the paper would make the results more reliable for informing the practical risk management strategies and other conclusions listed at the end of the paper.

**Reply**

We appreciate the learned reviewer for his keen interest in reviewing our manuscript. His insightful comments and constructive feedback has been helpful in enhancing the quality of the manuscript.

As suggested by the reviewer we have considered a few more scenarios and also took into account the stochastic variability in the control source parameters. The results and findings are discussed and shall be added to the paper.

We have taken three distinct seismic scenarios: i) the Car Nicobar earthquake of 1881, ii) the North Andaman earthquake of 1941 (Table 1 a, b), and iii) the Andaman Sumatra Subduction Zone earthquake of 2004, with a particular focus on the 2004 Sumatra tsunami earthquake, extensively discussed within our paper. For each scenario, we have computed the arrival times (Minutes), maximum wave height (m), and inundation (m). Plots depicting these computations

along the gauge locations in the southern Andaman region are provided: Fig 1a and b for the Car Nicobar earthquake of 1881, and Fig 1c and d for the North Andaman earthquake of 1941.

Table 1. Deformation parameters used to simulate different scenarios a) Car Nicobar, 1881 earthquake b) North Andaman, 1941 earthquake (Mishra et al. 2014)

| Earthquake parameters and source location | a) Car Nicobar, 1881 | b) North Andaman, 1941 |
|---|---|---|
| Latitude (E) | 8.52 | 12.1 |
| Longitude (N) | 92.43 | 92.5 |
| Data of occurrence | 31.12.1881 | 26.06.1941 |
| EQ magnitude (Mw) | 7.9 | 7.7 |
| Fault Length (km) | 200 | 200 |
| Fault width (km) | 80 | 80 |
| Strike (deg) | 350 | 20 |
| Slip (m) | 5 | 5 |
| Dip (deg) | 25 | 20 |
| Rake (deg) | 90 | 90 |
| Depth (km) | 15 | 30 |

[Figure]

Fig 1a directivity and b wave run-up height for the Car Nicobar earthquake of 1881, and Fig 1c and d for the North Andaman earthquake of 1941. (Andaman-Sumatra 2004 earthquake added in Manuscript)

Table 2. Estimated Run-up heights, Arrival Times, and inundations at the study region from i) Car Nicobar, 1881 earthquake ii) North Andaman, 1941 earthquake, and iii) Andaman Sumatra Subduction Zone, 2004 earthquake

| Location | Parameters | Earthquake Events | | |
|---|---|---|---|---|
| | | Carnicobar, 1881 | North Andaman, 1941 | Andaman Sumatra, 2004 |
| Rutland Island | Arrival Time (Min) | 26.55 | 25.9 | 27 |
| | Run up height (m) | 1.44 | 1.01 | 6 |
| | Inundation (m) | 50-380 | 10-220 | 700 |
| Wandoor | Arrival Time (Min) | 42.80 | 22.5 | 36.5 |
| | Run up height (m) | 2.21 | 1.25 | 3.5 |
| | Inundation (m) | 20-200 | 10-180 | 450 |
| Tirupati Temple, Port Blair | Arrival Time (Min) | 46.5 | 21.75 | 38 |
| | Run up height (m) | 1.65 | 1.42 | 1 |
| | Inundation (m) | 20-400 | 10-360 | 200 |
| Wimberlygunj | Arrival Time (Min) | 32.95 | 25.6 | 58 |
| | Run up height (m) | 1.40 | 1.12 | 12.5 |
| | Inundation (m) | 10-250 | 10-180 | 210 |
| Shoal Bay | Arrival Time (Min) | 42.5 | 34.8 | 56 |
| | Run up height (m) | 1.45 | 1.77 | 13 |
| | Inundation (m) | 20-220 | 10-180 | 950 |
| Chidiyatopu | Arrival Time (Min) | 26.5 | 21.75 | 36 |
| | Run up height (m) | 2.05 | 1.79 | 3.9 |
| | Inundation (m) | 20-500 | 20-300 | 585 |
| South Point, Port Blair | Arrival Time (Min) | 28.4 | 22 | 31.5 |
| | Run up height (m) | 2.31 | 2.12 | 9.6 |
| | Inundation (m) | 20-500 | 20-280 | 550 |
| | Arrival Time (Min) | 28.8 | 22.3 | 33 |

| Corbyns Cove Beach | Run up height (m) | 2.3 | 2.1 | 12.7 |
|---|---|---|---|---|
| | Inundation (m) | 20-580 | 10-320 | 900 |
| Bombooflat | Arrival Time (Min) | 31.2 | 24.55 | 42 |
| | Run up height (m) | 2.35 | 2.23 | 5.5 |
| | Inundation (m) | 20-650 | 20-350 | 90 |

Comparing three different scenarios, it is observed that the run-up height along the Eastern coast of South Andaman is greater than the Western coast (Fig. 1 c and d). This difference is due to the wider continental shelf on the Western coast of the south Andaman region and shallow water depths. In case of a higher magnitude of tsunamigenic earthquakes in the Car Nicobar or the North Andaman region, we would get higher run-ups along the gauge locations.

The arrival times of tsunamis varied across all locations, with each earthquake event exhibiting a range from approximately 21.75 minutes to 58 minutes (Table 2). The Car Nicobar, 1881 earthquake generally resulted in shorter arrival times compared to the other two events. Run-up heights ranged from approximately 1 meter to as high as 13 meters. Particularly, the Andaman Sumatra, 2004 earthquake tended to generate the highest run-up heights, in locations like Shoal Bay and Corbyns Cove Beach. The extent of inundation, representing the area covered by the tsunami, exhibited wide variation across locations and earthquake events, with depths ranging from 10 meters to a maximum of 950 meters. The Andaman Sumatra, 2004 earthquake led to the most extensive inundations, especially notable at locations like Shoal Bay and Corbyns Cove Beach. Comparing the three earthquake events, the Andaman Sumatra, 2004 earthquake generally resulted in higher run-up heights and more extensive inundations compared to the Car Nicobar, 1881, and North Andaman, 1941 earthquakes. The observed run-up heights and inundations were influenced by various factors, including the magnitude of the earthquake, the proximity of the source to observation locations, and the local coastal topography. Shoal Bay and Corbyns Cove Beach experienced significant inundations during the Andaman Sumatra, 2004 earthquake, highlighting the urgent need for robust mitigation and preparedness measures in these vulnerable coastal regions.

We acknowledge the importance of robust population tsunami safety studies and have taken steps to enhance our analysis accordingly. In addition to evaluating historical tsunami events, we have incorporated additional scenario analyses to capture the stochastic variability of tsunami wave heights and consider local upward variations. By expanding our study to include

an ensemble of alternative tsunami scenarios, particularly those impacting regions of recent economic development, we aim to provide more reliable insights for practical risk management strategies. This involves engaging with local communities, optimizing evacuation planning, enhancing early warning systems, fortifying infrastructure resilience, and adopting a multi-hazard risk assessment approach (National Research Council, 2011). Our study contributes to this broader goal by providing essential data and insights to support evidence-based decision-making and mitigate the adverse impacts of tsunamis on coastal population.

**References**

Mishra, P., Usha, T., & Ramanamurthy, M. V. (2014). Evaluation of tsunami vulnerability along northeast coast of India. Continental Shelf Research, 79, 16-22.

National Research Council (2011) Tsunami warning and preparedness: an assessment of the US tsunami program and the nation's preparedness efforts, Committee on the Review of the Tsunami Warning and Forecast System and Overview of the Nation's Tsunami Preparedness, National Research Council, 284 pp

---

## Author Comment (AC2)

**Response to Reviewer 2**

**Ms. No: Preprint nhess-2023-191**

**Title**: **Shoreline and Land Use Land Cover Changes along the 2004 tsunami-affected South Andaman Coast: Understanding Changing Hazard Susceptibility. https://doi.org/10.5194/nhess-2023-191**

**In this response file, the text in blue shows the comments from reviewers, while the black text is our replies.**

We highly appreciate the learned reviewer for his keen interest in reviewing our manuscript. His insightful comments and constructive feedback have helped enhance the quality of the manuscript. We have included most of the suggestions in the revised manuscript. Hope you find them appropriate.

**Reviewers comment**

**The author adopted the TUNAMI-N2 model to evaluate the area submerged by the tsunami flow. The authors should describe: i) the model, ii) the calibration parameters and how they are selected; and iii) the characteristics of the computational grid. The model is applied to a real event, therefore a validation with some field data could be useful.**

**Reply**

**The TUNAMI-N2 model and Computational grids**

The TUNAMI-N2 model is a numerical simulation tool widely employed for modeling tsunami propagation and inundation dynamics. Specifically, it utilizes finite-difference methods to solve the shallow water wave equations, incorporating factors such as bathymetry data, earthquake source parameters, and fault geometry to simulate tsunami behavior. The TUNAMI-N2 code utilizes input data organized into three columns: X-coordinate (Longitude), Y-coordinate (Latitude), and Z-values representing elevations (negative for land elevations and positive for ocean depths). These data are initially formatted and processed using Surfer software to convert into evenly spaced grids. We adopted a grid spacing ratio of 1:3 across all four grids. Grids A and B, designed to model linear effects in deep-sea regions, were set at resolutions of 81 arc seconds and 27 arc seconds, respectively. Conversely, grids C and D, aimed at capturing the non-linear effects of the tsunami, were maintained at constant

resolutions of 9 arc seconds and 3 arc seconds, respectively. As the tsunami wave propagates from deep waters to shallow waters the non-linear effects come into the picture i.e. amplitude dispersion, energy dissipation, bottom friction, and shallow depths. The program assumes non-linear theory to estimate the run-ups and impact. This program uses nesting of grids with accurate bathymetry and topography data to simulate the tsunami i.e. A, B, C, and D grids. We used coarser resolution bathymetry and topography data to model the initial deformation and propagation to save CPU time. Nested grids minimize errors by ensuring a sufficient number of nodes within each wavelength to accurately resolve the wave. In deep water, where wavelengths are longer, relatively coarse grids suffice to resolve the wave with minimal error, as fewer nodes are needed. However, as the wave propagates into shallower waters, the wavelength shortens and the amplitude increases. This necessitates finer resolution grids with more node points to accurately capture the wave dynamics and prevent errors. Therefore, higher-resolution grids are required in shallow waters to ensure an accurate representation of wave behavior. We considered the spacing of grids in such a way that it satisfies the Courant-Friedrich-Lewy conditions for checking the convergence of the numerical code to a certain asymptotic limit.

$$\Delta x/\Delta t = \sqrt{(2ghmax)}$$

Where $\Delta t$ and $\Delta x$ are temporal and spatial grid sizes, hmax maximum still water depth in the computational domain, and g is the gravitational acceleration. In our study, we considered spatial grid spacing by keeping the time step constant i.e. 15 seconds, and temporal grid spacing of 1 second.

Apart from the bathymetry and topography data, one should have precise focal mechanism solutions and fault parameters to compute the initial deformation at the source at t=0 seconds. This code uses Mansinha and Smyile's (1971) deformation model to estimate the seafloor upliftment near the source. TUNAMI-N2 code allows the tsunami wave to propagate freely in the open sea and for that, we have to consider the exterior grid (A) in a very large domain as the tsunami propagates transoceanic regions and is interpolated into the B, C, and D grids. After giving the required inputs the program is compiled and executed to get the directivity map, wave amplitudes at different tide-gauge locations, and run-up heights at different locations in the study region.

**Calibration Parameters and how they select:**

The calibration parameters include bathymetry resolution, earthquake source parameters (e.g., slip distribution, fault length, and width), and grid resolution. These parameters were carefully

selected based on established literature (e.g., Ioualalen, 2007; Rani et al., 2011; Srivastava et al., 2021) and sensitivity analyses to ensure their appropriateness for capturing the characteristics of the 2004 Sumatra earthquake and subsequent tsunami.

**Validation with Field Data:**

We have validated our model results with field data as shown in Figure 3 of the manuscript. We also used the observations of Cho et al. (2008) and Prerna et al. (2015) to highlight the consistency of our findings with respective field studies and we have visited several coastal locations in their study. The agreement underscores the reliability of our numerical simulations.

**Regarding the shoreline changes, uncertainty must be evaluated. Due to the low slope of the beach in some transects, uncertainty must be correlated with the water level (tide and barotropic surge).**

The evaluation of uncertainty encompasses various factors, including natural and anthropogenic forces such as wind, waves, tides, currents, and human influences, along with the accuracy of measurement techniques, including digitization, interpretation, and GPS error. Special attention has been given to the influence of tides on Landsat satellite imagery used in shoreline analysis. Even though, because of the large coastline extent of the study area, the tidal difference would only be visible to an inconsiderable amount.

As tide gauge data for Port Blair station is unavailable for the years 2004-2005 and 2022, we have calculated uncertainty using available data from 2018-2019 and 2019-2020 obtained from the Permanent Service for Mean Sea Level (PSMSL) database (https://psmsl.org/data/obtaining/stations/206.php). The calculated uncertainty values for these years are 7.46 meters and 7.13 meters, respectively. To quantify uncertainty, we have adopted a confidence interval of 90% and assigned a shoreline uncertainty value of 10 meters. This aligns with recommendations from the U.S. Geological Survey (USGS), which suggests a default value of 10 meters based on recent regional reports under the National Assessment of Shoreline Change project (Himmelstoss et al., 2021; Den and Oele, 2018 and Joesidawati, 2016).

We have incorporated this aspects in the revised manuscript as well.

**NSM and EPR are not "statistical" parameters, since they are related to the difference between two observations.**

We agree that EPR (End Point Rate) and NSM (Net Shoreline Movement) are not statistical parameters. EPR and NSM are not considered statistical parameters because they do not involve the estimation of parameters based on sample data or statistical inference. Instead, they are quantitative measures derived from observed data (i.e., shoreline positions) over specific time intervals. They represent the calculated rates of shoreline change and are not inherently statistical in nature. They are used to quantify and characterize the rates of erosion or accretion along coastlines, providing valuable information for understanding coastal dynamics and assessing coastal hazards. (Himmelstoss et al, 2021; Sam and Gurugnanam, 2022; Den and Oele, 2018; Ciritci and Türk, 2020).

However, mean values of these parameters have been computed and we mentioned them as statistical parameter primarily based on Himmelstoss et al., 2021, where it is referred as Statistical Parameter.

We have modify the text in revised manuscript to clear this ambiguity.

**Minor points:**

**1. L. 48-50 – check the sentence.**

Reply: Sentence corrected in the revised manuscript.

**2. Figure 3 a and b – please add labels in the axes and the colour bars.**

Reply: Labels axes is now added and the colour bar is already present in Figures 3 a and b

[Figure]

**3. L, 226 – Delete "rates". EPR is already a rate.**

Reply: "rate" is now deleted

**4. Figure 5 – the axis labels are too small.**

Reply: Axis label size is now increases

[Figure]

**5. pages 14-15 – Check the reference to figures SM1 – SM4,**

Reply: SM1 – SM7 is now changed to S1, S2…S7

**6.L. 285-288. Are you sure about the change in water depth? The ground colors in the 2005 and 2022 images also show a noticeable difference.**

The dark blue color in the Landsat images from 2004 and 2005 suggests clear water without detrital sediment load, while the light blue color in the 2022 image indicates a significant fresh sediment load with bright reflectance and we assume that it will have effect on reduction in water column depth.

**References**

Himmelstoss, E., Henderson, R. E., Kratzmann, M. G., & Farris, A. S. (2021). Digital shoreline analysis system (DSAS) version 5.1 user guide (No. 2021-1091). US Geological Survey.

Sam, S. C., & Gurugnanam, B. (2022). Coastal transgression and regression from 1980 to 2020 and shoreline forecasting for 2030 and 2040, using DSAS along the southern coastal tip of Peninsular India. Geodesy and Geodynamics, 13(6), 585-594.

Den Boer, E. L., & Oele, A. C. (2018). Determination of shoreline change along the East-Java coast, using the Digital Shoreline Analysis System. In *MATEC Web of Conferences* (Vol. 177, p. 01022). EDP Sciences.

Ciritci, D., & Türk, T. A. R. I. K. (2020). Assessment of the Kalman filter-based future shoreline prediction method. International journal of environmental science and technology, 17(8), 3801-3816.

Joesidawati, M. I. (2016). Shoreline change in Tuban district, East Java using geospatial and digital shoreline analysis system (DSAS) techniques. International Journal of Oceans and Oceanography, 10(2), 235-246.

Mansinha, L., and Smylie, D.E., 1971, The displacements fields of inclined faults, Bull. Seismol. Soci. Am., 61(5), 1433-1440.

---

## Author Response (AR1)

**NATIONAL GEOPHYSICAL RESEARCH INSTITUTE**
**Council of Scientific & Industrial Research**
**Uppal Road, Hyderabad – 500007, India**

**Dr. Anand K. Pandey**
*Chief Scientist,*
*Professor (Academy of Scientific Innovation & Research-AcSIR)*
*Head, Geology Group*

May 8, 2024

**Subject**: Submission of the revised manuscript (**Ms. No: NHESS-2023-191**)

Dear Dr. Brunella Bonaccorso,

Thanks for the mail informing "major revision" of our manuscript NHESS-2023-191 based on comments of learned reviewers. The comments of the reviewers have been insightful and helped in revising the manuscript with enhance clarity and robustness. We have simulated a few scenarios and incorporated them in the revised manuscript. Please find the same as attachment.

In light of insightful comments, we have carried out additional tsunami scenarios for the 1881-Car Nicobar, 1941-North Andaman earthquake in addition to the 2004 Sumatra earthquake. Since the latter has affected the region with highest severity, it has been considered for the scenario tsunami risk analysis (Sections 3.2 and 4.1). A brief description of the TUNAMI-N2 model, including calibration parameters and computational grid has also been added to the methodology section 3.2 as per the suggestions. The shoreline uncertainty evaluation and minor points raised by the reviewer regarding figure quality, language corrections, and references have been carefully addressed. The detailed response to the respective reviewers are in the following pages. Needless to mention we are attaching the "clean" and "Track-change" versions of the manuscript.

We hope the revised manuscript is significantly improved and would make important contribution to the field of natural hazards and disaster management and would find audition in the Natural Hazards and Earth System Sciences journal.

We look forward to your kind response.

With sincere regards,

(Anand K. Pandey)
* * *
Email: akpandey@ngri.res.in          ☏ +91-40-27012416          Fax: +91-40-23434651
        akpngri@gmail.com          Website: www.ngri.org.in          Mobile +91-9440317602

[Figure]

**NATIONAL GEOPHYSICAL RESEARCH INSTITUTE**
**Council of Scientific & Industrial Research**
**Uppal Road, Hyderabad – 500007, India**

**Dr. Anand K. Pandey**
*Chief Scientist,*
*Professor (Academy of Scientific Innovation & Research-AcSIR)*
*Head, Geology Group*

**Our Reply** to the COMMENTS of REVIEWERS on Ms. No: NHESS-2023-191

The reply is marked in color.

**Title**: **Shoreline and Land Use Land Cover Changes along the 2004 tsunami-affected South Andaman Coast: Understanding Changing Hazard Susceptibility.**
**https://doi.org/10.5194/nhess-2023-191**

**Reply to Reviewer #1**

This paper addresses an important tsunami risk problem, which is aggravated by the increasing trend in population exposure, tourism etc. The analysis undertaken identifies a number of important tsunami risk issues, but falls short of what is required for a robust population tsunami safety study. To make the conclusions more robust, some additional scenario analyses would be insightful and instructive.

We highly appreciate the learned reviewer for his keen interest in reviewing the manuscript and providing insightful comments and constructive feedback that has helped enhance the quality of the manuscript.

First, all tsunami wave height outcomes are subject to a substantial degree of stochastic variability. Venturing beyond the actual 2004 tsunami wave height measurements, the implications for local upward variations in wave height should be considered by perturbing the tsunami source dynamics. Furthermore, other potentially dangerous earthquake-generated tsunamis merit attention, and an ensemble of some alternative potential tsunami scenarios should be considered, especially those that impact regions of recent economic development. This broadening of the basic tsunami modeling content of the paper would make the results more reliable for informing the practical risk management strategies and other conclusions listed at the end of the paper.

As suggested we have considered 3 Tsunamigenic scenarios namely, a) the 1881-Car Nicobar earthquake, b) the 1941-North Andaman earthquake, and c) the 2004 Sumatra earthquake, and generated the directivity and run-up map (Fig. 3; Table 2). The results and findings are discussed in sections 3.2 and 4.1.

Since the tsunami by the 2004 Sumatra earthquake has affected the region with highest severity, it has been considered for the scenario tsunami risk analysis (Sections 4.2).

We agree that analysis for each event is important but would be beyond the scope of present manuscript.

I hope the learned reviewer will find the revised manuscript appropriate.

Email: akpandey@ngri.res.in    ☎ +91-40-27012416    Fax: +91-40-23434651
akpngri@gmail.com    Website: www.ngri.org.in    Mobile +91-9440317602

[Figure]

**NATIONAL GEOPHYSICAL RESEARCH INSTITUTE**
**Council of Scientific & Industrial Research**
**Uppal Road, Hyderabad – 500007, India**

**Dr. Anand K. Pandey**
*Chief Scientist,*
*Professor (Academy of Scientific Innovation & Research-AcSIR)*
*Head, Geology Group*

**Reply to Reviewer #2**

At the outset we highly appreciate the learned reviewer for his providing insightful and constructive comments that has helped enhance the quality of analysis in the manuscript.

The author adopted the TUNAMI-N2 model to evaluate the area submerged by the tsunami flow. The authors should describe: i) the model, ii) the calibration parameters and how they are selected; and iii) the characteristics of the computational grid. The model is applied to a real event, therefore a validation with some field data could be useful.

We agree to the important suggestion for the completeness of the manuscript and have incorporated a short text addressing the above points into the methodology section 3.2

Regarding the shoreline changes, uncertainty must be evaluated. Due to the low slope of the beach in some transects, uncertainty must be correlated with the water level (tide and barotropic surge).

We have adopted a confidence interval of 90% and assigned a shoreline uncertainty value of 10 meters as per the recommendations of the United States Geological Survey (USGS) under the National Assessment of Shoreline Change project (Himmelstoss et al., 2021; Den and Oele, 2018 and Joesidawati, 2016). The short discussion about uncertainty is incorporated in sections 3.3

NSM and EPR are not "statistical" parameters, since they are related to the difference between two observations.

Mean values of these parameters have been computed and we mentioned them as statistical parameters primarily based on Himmelstoss et al., 2021, where it is referred to as Statistical Parameters.

**Minor points:**

1. L. 48-50 – check the sentence.

   Sentence corrected in the revised manuscript.

2. Figure 3 a and b – please add labels in the axes and the color bars.

   We have redrafted Fig.3 incorporating suggestions.

3. L, 226 – Delete "rates". EPR is already a rate.

   "rate" is now deleted

4. Figure 5 – the axis labels are too small.

   We have redrafted Fig. 6 incorporating suggestions.

5. pages 14-15 – Check the reference to figures SM1 – SM4,

   SM1 – SM7 is now changed to S1, S2…S7

Email: akpandey@ngri.res.in     ☎+91-40-27012416     Fax: +91-40-23434651
        akpngri@gmail.com     Website: www.ngri.org.in     Mobile +91-9440317602

[Figure]

**NATIONAL GEOPHYSICAL RESEARCH INSTITUTE**
**Council of Scientific & Industrial Research**
**Uppal Road, Hyderabad – 500007, India**

**Dr. Anand K. Pandey**
*Chief Scientist,*
*Professor (Academy of Scientific Innovation & Research-AcSIR)*
*Head, Geology Group*

6.L. 285-288. Are you sure about the change in water depth? The ground colors in the 2005 and 2022 images also shows a noticeable difference.

The dark blue color in the Landsat images from 2004 and 2005 suggests clear water without detrital sediment load, while the light blue color in the 2022 image indicates a significant fresh sediment load with bright reflectance, and we assume that it will affect the reduction in water column depth.

I hope the learned reviewer will find the revised manuscript appropriate.

Email: akpandey@ngri.res.in  ☎ +91-40-27012416  Fax: +91-40-23434651
akpngri@gmail.com  Website: www.ngri.org.in  Mobile +91-9440317602

---

## Author Response (AR2)

**NATIONAL GEOPHYSICAL RESEARCH INSTITUTE**
**Council of Scientific & Industrial Research**
Uppal Road, Hyderabad – 500007, India

**Dr. Anand K. Pandey**
*Chief Scientist,*
*Professor (Academy of Scientific Innovation & Research-AcSIR)*
*Head, Geology Group*

June 26, 2024

**Subject**: Submission of the revised manuscript (**Ms. No: NHESS-2023-191**)

Dear Dr. Brunella Bonaccorso,

Thanks for the mail communicating insightful comments of learned reviewers. We appreciate the additional suggestions of Reviewer #2. We carefully addressed the specific issues, and our response is detailed under,

As suggested by the reviewer, we have added the detailed description of the TUNAMI-N2 model in the text. The computational grid characteristics (A-D grids) and their roles in modeling are explained and added in supplementary material. Validation with field data shows our results align within a 10% error margin with observed tsunami run-up heights. The other queries regarding evaluating uncertainties, including tidal excursion, lack of barotropic surge consideration, and the mean slope in the study area, have also been explained. The shoreline positional error and its components, along with the calculation of total uncertainties, are explained in the supplementary materials. The other minor corrections in language and updated references are included in the revised manuscript. The "reply to review" also presents the details response.

As suggested, the revised submission includes the "clean" and "Track-change" versions. We hope you find the revised manuscript addressing the issues raised by Reviewer#2 appropriately and favorably consider the manuscript for publication.

We look forward to your kind response.

With sincere regards,

(Anand K. Pandey)

Email: akpandey@ngri.res.in    ☎ +91-40-27012416    Fax: +91-40-23434651
akpngri@gmail.com    Website: www.ngri.org.in    Mobile +91-9440317602

Title: Shoreline and Land Use Land Cover Changes along the 2004 tsunami-affected South Andaman Coast: Understanding Changing Hazard Susceptibility.
https://doi.org/10.5194/nhess-2023-191

In this response file, the blue text shows the reviewers' comments, while the black text is our replies.

Reply to Reviewer #2

The authors did not adequately reply to the following previous issues:

1) The author adopted the TUNAMI-N2 model to evaluate the area submerged by the tsunami flow. The authors should describe: i) the model, ii) the calibration parameters and how they are selected; iii) the characteristics of the computational grid. The model is applied to a real event, therefore a validation with some field data could be useful.

Thanks for the detailed comments.

We have incorporated a brief about the TUNAMI-N2 modeling, grid areas A-D, model input and calibration factors and validation, etc., in the revised manuscript. New graphs and figures are added to the supplementary materials. The details of the same are in the reply as under.

I hope you find the revision adequate and satisfactory.

- No relevant description of the TUNAMI-N2 model is provided (i.e. What equations does it solve? Is the solver implicit or explicit? …)

The Tohoku University's Numerical Analysis Model for the Investigation of Near field tsunamis (TUNAMI-N2) to simulate the tsunami run-ups and impact using explicit leap-frog finite-difference methods by solving non-linear shallow water wave equations, incorporating bathymetry, earthquake source parameters, and fault geometry (Imamura and Imteaz, 1995; Imamura, 1996; Goto, 1996; Imamura et al., 2006; Yalciner et al., 2003). The 2-dimensional governing equations for tsunami modeling are:

$$\frac{\partial \eta}{\partial t} + \frac{\partial M}{\partial x} + \frac{\partial N}{\partial y} = 0$$

$$\frac{\partial M}{\partial t} + \frac{\partial}{\partial x}\left(\frac{M^2}{D}\right) + \frac{\partial}{\partial y}\left(\frac{MN}{D}\right) + gD\frac{\partial \eta}{\partial x} + \frac{gn^2}{D^{7/3}}M\sqrt{M^2 + N^2} = 0$$

$$\frac{\partial N}{\partial t} + \frac{\partial}{\partial x}\left(\frac{MN}{D}\right) + \frac{\partial}{\partial y}\left(\frac{N^2}{D}\right) + gD\frac{\partial \eta}{\partial y} + \frac{gn^2}{D^{7/3}}N\sqrt{M^2 + N^2} = 0 \qquad (1)$$

where D is the total water depth given by h+η, τx, and τy the bottom frictions in the x- and y-directions, A is the horizontal eddy viscosity, which is a constant in space, and the shear stress on a

surface wave is neglected. M and N are the discharge fluxes in the x- and y- directions, which are given by

$$M = \int_{-h}^{\eta} u \, dz = u(h + \eta) = uD \qquad\qquad N = \int_{-h}^{\eta} v \, dz = v(h + \eta) = vD \qquad (2)$$

The bottom friction is generally expressed as follows

$$\frac{\tau_x}{\rho} = \frac{1}{2g}\frac{f}{D^2} M\sqrt{(M^2 + N^2)} \qquad\qquad \frac{\tau_y}{\rho} = \frac{1}{2g}\frac{f}{D^2} N\sqrt{(M^2 + N^2)} \qquad (3)$$

The friction coefficient 'f' and Manning's roughness 'n' are related by

$$n = \sqrt{\frac{f D^{1/3}}{2g}} \qquad\qquad (4)$$

It is seen that when D is small and f becomes large, then n remains almost a constant; substituting M, N, and the above values in fundamental equations of TUNAMI N2 to obtain wave propagation using the explicit Leap-Frog finite difference Scheme (Imamura, 2006).

- What are the areas A-D indicated on line 151?

TUNAMI-N2 code uses the data, which is formatted into three columns, X-coordinate (Longitude), Y-coordinate (Latitude), and Z (Land elevations as negative and Ocean depths as positive), and converted into evenly spaced grids by using surfer software. In our study, we considered grid spacing of all four grids in 1: 3 ratios, i.e., A and B grids to model the linear effects in the deep sea of 81 arc seconds and 27 arc seconds, and C and D constant grids to model the non-linear effects of the tsunami are of 9 arc seconds and 3-arc seconds. The A, B, C, and D grids are used to compute Tsunami wave height and inundation and are included in the supplementary materials as Figure (S1).

Email: akpandey@ngri.res.in          ☎ +91-40-27012416          Fax: +91-40-23434651
akpngri@gmail.com          Website: www.ngri.org.in          Mobile +91-9440317602

[Figure]

Figure S1: A-D grids used for the TUNAMI-N2 modeling in the present study

- The model input data are indicated, not the model calibration parameters (e.g., roughness, turbulence,..)

In most computations, the manning coefficient is around 0.025, consisting of gravel and sand (Masaya et al., 2020); however, different manning coefficients can be considered for rough bathymetry (Dao and Tkalich, 2007). A value of 0.01 is considered for smooth bathymetry and stony cobbles, and a roughness of 0.035 can be considered. Viscosity and roughness influence mild slopes, but they are negligible for steep slopes, and a dynamic friction coefficient from 0.01 to 0.1 can be considered (Zhang et al., 2024). For the propagation of tsunamis in shallow water, the horizontal eddy turbulence terms are negligible as compared with the bottom friction (Dao and Tkalich, 2007)

- No sort of validation is provided with data relating to the events analysed

Our results agree with the tsunami run-up heights estimation by Cho et al. (2008) and Prerna et al. (2015) at a few locations in the present study area. Since the tide gauge data are available at a few locations along the Indian coast, we rely on limited field observations along the coast to validate our

Email: akpandey@ngri.res.in     ☎ +91-40-27012416     Fax: +91-40-23434651
akpngri@gmail.com     Website: www.ngri.org.in     Mobile +91-9440317602

findings. The field observations of the water marks on a light post at Bambooflat in Port Blair was seen to be around 3.8m (Cho et al., 2008) and our computations show it to be ~ 3.5m, which is within ~7% error limit. Similarly, at South Point, Port Blair, the field observations are 10m, and our computations value is 9.6m, which is ~4% deviation and the deviation is 7% at Chidiyatopu. The Bambooflat region and Harbour area of Port Blair experienced liquefaction affecting several buildings (Murty et al., 2006), and our calculations show that the tsunami wave heights were around 5.5m. At most locations, the computed values are within 10% error.

2. L. 186-187 How did you evaluate the uncertainty? What is the measured tide excursion? What is the barotropic surge? What is the mean slope in the area?

The accuracy of shoreline position and the rates of shoreline change can be influenced by various error sources, such as the position of the tidal level, image resolution, digitization error, and image registration (Jayson-Quashigah et al., 2013; Vu et al., 2020, Basheer et al., 2022). Therefore, the shoreline positional error (Ea) for each transect was calculated using Equation (6):

$$E_s = \pm\sqrt{E_s^2 + E_w^2 + E_d^2 + E_r^2 + E_p^2} \qquad (6)$$

Where Es is the seasonal error due to seasonal shoreline fluctuations, which is ~ ±5 m in extreme ocean level (EOL); Ew is the tidal error, Ed is the digitization error, Er is the rectification error, and Ep is the pixel error (Fletcher et al. 2011; Vu et al., 2021). This approach assumes that the component errors are normally distributed (Dar & Dar, 2009). The total uncertainties were used as weights in the shoreline change calculations. The values were annualized to provide errors (Eu) estimation for the shoreline change rate at any given transect, expressed in Equation (7):

$$E_u = \pm\frac{\sqrt{U_{t1}^2 + U_{t2}^2 + U_{t3}^2 + U_{t4}^2 + U_{tn}^2}}{T} \qquad (7)$$

where $t_1$, $t_2$, and tn are the total shoreline position error for the various years, and T is the years of analysis.

Email: akpandey@ngri.res.in     ☎ +91-40-27012416     Fax: +91-40-23434651
akpngri@gmail.com     Website: www.ngri.org.in     Mobile +91-9440317602

Table S1 Uncertainty Calculation used in the DSAS tool

| Positional Error (m) | 2003 | 2004 | 2017 | 2018 | 2019 | 2020 |
|---|---|---|---|---|---|---|
| Seasonal error (Es)) | 5 | 5 | 5 | 5 | 5 | 5 |
| Tidal fluctuation ($E_{td}$) | 1.17 | 0.38 | 0.86 | 1.1 | 0.84 | 0.85 |
| Shoreline proxy offset (Eo) | NA | NA | NA | NA | NA | NA |
| Measurement errors (m) | 0 | 0 | 0 | 0 | 0 | 0 |
| Georeferecing/Rectification error (Er) | 0 | 0 | 0 | 0 | 0 | 0 |
| Digitizing error (Ed) | 20 | 20 | 15 | 20 | 20 | 19 |
| Toposheet survey offset (Et) | NA | NA | NA | NA | NA | NA |
| Pixel Error (Ep) | 0 | 0 | 0 | 0 | 0 | 0 |
| Total Shoreline position error (Esp) m | 26.17 | 25.38 | 20.86 | 26.10 | 25.84 | 24.85 |
|  |  |  |  |  |  |  |
| Year | 2003-04 | 2004-05 | 2017-18 | 2018-19 | 2019-2020 | 2020-2021 |
| Uncertainty | 7.18 | 5.04 | 6.85 | 7.21 | 7.12 | 4.98 |

- What is the measured tide excursion?

  Highest Tide Gauge Measurement: 1100 mm
  Lowest Tide Gauge Measurement: 717 mm
  Tide Excursion: 1100 - 717 = 383 mm
  The measured tide excursion from 2017 to 2020 is 383 mm (0.383 m).

[Figure]

Figure S2 Monthly mean sea level data showing tide excursion (Tide Gauge data)

**What is the barotropic surge?**

Our current analysis focused on shoreline change detection due to tsunamis and decades after tsunamis using Landsat-8 data and the DSAS tool. The observed tidal effect is used for error analysis of inundation due to the 2004 tsunami. We did not incorporate barotropic surge by large-scale

Email: akpandey@ngri.res.in    ✆ +91-40-27012416    Fax: +91-40-23434651
akpngri@gmail.com    Website: www.ngri.org.in    Mobile +91-9440317602

atmospheric pressure variations, which can cause temporary water level fluctuations and, in extreme cases, might influence shoreline change.

Including surge data could be valuable for future investigations aiming at a comprehensive understanding of shoreline dynamics in the study region.

**What is the mean slope in the area?**

The mean slope of an area is a crucial topographic characteristic that influences various coastal processes, including erosion, sediment transport, and shoreline stability. We digitized the shoreline along seven zones and marked a 500-meter buffer to check the mean slope along the zones. The mean slope values for these Zones are derived from a DEM using a zonal statistics tool in the ArcGIS environment.

Table S2 Mean Slope of Area

| Zone | AREA (Sq. m) | MIN (Degrees) | MAX (Degrees) | RANGE (Degrees) | MEAN (Degrees) | STD (Degrees) |
|------|--------------|---------------|---------------|-----------------|----------------|---------------|
| 1 | 9775286.671 | 0 | 36.98 | 36.98 | 7.22 | 4.69 |
| 2 | 4079097.873 | 0 | 24.96 | 24.96 | 8.18 | 4.66 |
| 3 | 2443355.379 | 0 | 23.50 | 23.50 | 6.60 | 3.95 |
| 4 | 2830375.41 | 0 | 20.72 | 20.72 | 4.13 | 3.32 |
| 5 | 2875139.173 | 0 | 33.82 | 33.82 | 5.56 | 4.52 |
| 6 | 1612428.034 | 0 | 33.75 | 33.41 | 11.4 | 6.90 |
| 7 | 6126107.436 | 0 | 37.81 | 37.81 | 12.3 | 5.99 |

[Figure]

Figure S3 Slope Map showing 500m buffer marked along the shoreline

Email: akpandey@ngri.res.in          ☎ +91-40-27012416          Fax: +91-40-23434651
       akpngri@gmail.com          Website: www.ngri.org.in          Mobile +91-9440317602

3) Again, NSM and EPR are not "statistical" parameters since they relate to the difference between two observations.
We have incorporated your suggestion.

4) L. 342-345 It's just an opinion and not a fact. The presence of suspended sediment do not is directly associated with a reduction in the water depth.
Agree and acknowledge the limitations of solely relying on visual interpretation. However, the increased suspended sediment load derived from land use change would contribute to shoaling (shallowing of water bodies) over time, especially in bay areas.

**References for the reply (added to the Ms_R2)**

Davis, R.A.:Human Impact on Coasts. In: Finkl, C.W., Makowski, C. (eds) Encyclopedia of Coastal Science. Encyclopedia of Earth Sciences Series. Springer, Cham, https://doi.org/10.1007/978-3-319-93806-6_175, 2019.

Basheer Ahammed, K. K., & Pandey, A. C. (2022). Assessment and prediction of shoreline change using multi-temporal satellite data and geostatistics: A case study on the eastern coast of India. Journal of Water and Climate Change, 13(3), 1477-1493.

Fletcher, C.H., Romine, B.M., Genz, A.S., Barbee, M.M., Dyer, M., Anderson, T.R., Lim, S.C., Vitousek, S., Bochicchio, C., & Richmond, B.M. (2011). National assessment of shoreline change: Historical shoreline change in the Hawaiian Islands. In (p. 55).

Vu, M. T., Lacroix, Y., & Vu, Q. H. (2021). Assessment of the Shoreline Evolution at the Eastern Giens Tombolo of France. In Proceedings of the International Conference on Innovations for Sustainable and Responsible Mining: ISRM 2020-Volume 2 (pp. 349-372). Springer International Publishing.

Murty, C. V. R., Rai, D. C., Jain, S. K., Kaushik, H. B., Mondal, G., & Dash, S. R. (2006). Performance of structures in the Andaman and Nicobar Islands (India) during the December 2004 great Sumatra earthquake and Indian Ocean tsunami. Earthquake spectra, 22(3_suppl), 321-354.

Masaya, R., Suppasri, A., Yamashita, K., Imamura, F., Gouramanis, C., & Leelawat, N. (2020). Investigating beach erosion related with tsunami sediment transport at Phra Thong Island, Thailand, caused by the 2004 Indian Ocean tsunami. Natural Hazards and Earth System Sciences, 20(10), 2823-2841.

Dao, M. H., & Tkalich, P. (2007). Tsunami propagation modeling–a sensitivity study. Natural Hazards and Earth System Sciences, 7(6), 741-754.
* * *
The part of the above explanations is included in the text of the manuscript and figures in the Supplementary materials.

We hope reviewer#2 finds the explanations and modifications in the Ms appropriate.

Anand

--

Email: akpandey@ngri.res.in ☎ +91-40-27012416 Fax: +91-40-23434651
akpngri@gmail.com Website: www.ngri.org.in Mobile +91-9440317602

---

## Author Response (AR3)

**Author's Response**

Thanks for accepting the manuscript.

As suggested, we have made the following corrections to the text

- Reduced the decimal digits to two places and converted the unit from $m^2$ to $Km^2$ in Table S2.
- Added 'Google Earth' to the credit in Fig S1.
- Updated the reference list and added Doi.